# FlowPET: Physics-Informed Symplectic Flow Matching for Low-Count PET Reconstruction

**Zheng Zhang** [1] **Hao Tang** [1] **Yingying Hu** [2] **Zhanli Hu** [3] **Jing Qin** [1]

## Abstract

Low-count Positron Emission Tomography (PET) reconstruction is severely hindered by the dissipative nature of prevailing generative models, where the inherent phase-space contraction leads to the numerical extinction ("wash-out") of weak but diagnostically critical lesion signals. To overcome this geometric limitation, we propose **FlowPET**, a physics-informed framework that reformulates reconstruction as volume-preserving transport in a symplectic phase space. By parameterizing the posterior dynamics via a Separable Hamiltonian System, our approach guarantees a divergence-free vector field by construction, theoretically immunizing weak signals against probability mass collapse. To steer this conservative flow, we introduce conjugate boundary conditions based on the Range-Null space decomposition of the PET operator; this strictly enforces data consistency in the range space while confining stochastic uncertainty injection to the unobserved null space. We train the model via symplectic flow matching and perform inference using a symplectic leapfrog integrator. Extensive experiments on BrainWeb, clinical pediatric, and UDPET datasets demonstrate that **FlowPET** not only surpasses state-of-the-art deterministic and stochastic baselines in SSIM and PSNR but, more crucially, exhibits superior recovery of low-contrast lesions. The results confirm that imposing Hamiltonian structural constraints offers a robust geometric safeguard for medical inverse problems in high-noise regimes.

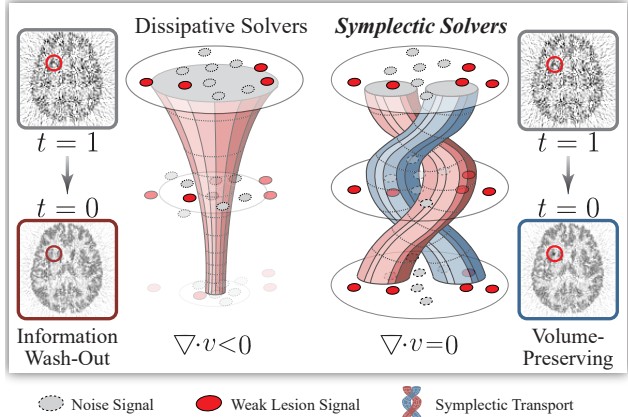

*Figure 1.* Conceptual comparison of generative dynamics in PET reconstruction. Standard dissipative solvers induce a contractive field ($\nabla \cdot v < 0$), causing weak lesion signals to be "washed out" alongside noise. In contrast, symplectic solvers lift the dynamics to a symplectic phase space, enforcing volume preservation ($\nabla \cdot v = 0$) to losslessly transport clinically vital signals under strict physical constraints.

## 1. Introduction

Low-count Positron Emission Tomography (PET) reconstruction represents a critical frontier in medical imaging, balancing the need for precise metabolic quantification against stringent radiation safety protocols. Reducing tracer dose inevitably degrades the Signal-to-Noise Ratio (SNR) due to Poisson photon-counting statistics. In this regime, the reconstruction task transcends simple inversion; it becomes a complex Bayesian inference problem aiming to recover the posterior distribution conditional on noisy measurements. The paramount challenge is not merely denoising, but the faithful recovery of *weak lesion signals*, features that are diagnostically vital for early staging yet statistically fragile against the dominant noise floor.

To mitigate the noise amplification inherent in classical algorithms (e.g., OSEM (Hudson & Larkin, 1994)), the field has pivoted to deep learning. While effective, current paradigms face a fundamental dilemma. Deterministic approaches (Zhang et al., 2026; 2024; Xie et al., 2025; Wang & Liu, 2020; Cui et al., 2024) minimize reconstruction error by approximating the posterior mean. This leads to the well-known "regression-to-the-mean" phenomenon, where high-

---

[1]Centre for Smart Health, The Hong Kong Polytechnic University, Hong Kong, China [2]Department of Nuclear Medicine, Sun Yat-sen University Cancer Center, Guangzhou, China [3]Research Center for Medical AI, Shenzhen Institute of Advanced Technology, Chinese Academy of Sciences, Shenzhen, China. Correspondence to: Hao Tang <howard.haotang@gmail.com>.

*Proceedings of the 43rd International Conference on Machine Learning*, Seoul, South Korea. PMLR 306, 2026. Copyright 2026 by the author(s).

frequency textures and subtle lesion contrasts are obliterated by statistical averaging. Generative paradigms, conversely, aim to restore detailed distributions. While Stochastic Differential Equation (SDE) solvers (Ai et al., 2025; Huang et al., 2025; Luo et al., 2023) significantly enhance textural realism, they fundamentally rely on *dissipative dynamics* to drive sampling trajectories. We identify a critical geometric flaw in this mechanism: standard diffusion flows induce a strictly negative divergence ($\nabla \cdot v < 0$), forcing a continuous contraction of the phase-space volume. For low-count PET, this contraction is indiscriminate; it fails to distinguish between stochastic noise and sparse pathological features. Consequently, weak lesion signals are prone to *numerical extinction*, as they are effectively "washed out" by the flow before forming coherent structures (Figure 1).

Preserving these fragile signals requires shifting from dissipative contraction to a conservative geometric framework. We introduce **FlowPET**, which reformulates posterior sampling as an invertible, volume-preserving evolution. Unlike standard methods that operate on the image manifold, we *lift* the reconstruction to a *Symplectic Phase Space*. By parameterizing the vector field via *Separable Hamiltonian Dynamics* within a *Flow Matching* framework, our method strictly obeys Liouville's Theorem (Safko et al., 2002). This allows the model to leverage the auxiliary momentum space to reorganize information without compression, ensuring that the probability mass of weak lesions is isometrically transported rather than extinguished.

However, volume preservation presents a new challenge: without the "denoising" effect of contraction, the model lacks an intrinsic mechanism to suppress noise. To navigate this, we must explicitly guide the conservative flow using the inherent anisotropy of the PET inverse problem. We enforce Physics-Informed Conjugate Boundaries based on the Range-Null space decomposition. Specifically, we encode data consistency gradients into the momentum variable (Range space) while confining stochastic uncertainty injection strictly to the orthogonal subspace (Null space). This design ensures that the generative process synthesizes plausible high-frequency textures without hallucinating artifacts that violate the underlying physics.

In summary, we make three primary contributions:

- **Symplectic Generative Framework:** We propose **FlowPET**, a framework utilizing separable Hamiltonian dynamics to ensure divergence-free transport. This theoretically prevents the "signal wash-out" pervasive in dissipative models by maintaining phase-space volume invariant.

- **Physics-Informed Orthogonality:** We introduce a Range-Null space decomposition strategy for phase-space boundaries. This rigorously enforces data con-

sistency in the range space while confining stochastic exploration to the null space, resolving the conflict between fidelity and diversity.

- **Structure-Preserving Inference:** We implement a symplectic Leapfrog integrator for inference, demonstrating that strict geometric conservation yields superior preservation of low-contrast lesions compared to state-of-the-art baselines on diverse low-count datasets.

## 2. Related Work

**PET Reconstruction** The trajectory of deep learning in PET reconstruction reflects a fundamental shift from deterministic regression to stochastic generative modeling. Early deterministic approaches primarily learn a mapping from low-count sinograms to high-quality images in a direct (Wang & Liu, 2020; Kaviani et al., 2023; Cui et al., 2024) or an iterative manners (Zhang et al., 2026; 2024; Xie et al., 2025; Hu & Liu, 2022; Fu et al., 2021). While these regression-based methods surpass classical iterative algorithms (Hudson & Larkin, 1994; Shepp & Vardi, 2007) in terms of peak SNR, they mathematically approximate the *posterior mean* $\mathbb{E}[x|y]$. Consequently, they are plagued by the "regression-to-the-mean" phenomenon (Saharia et al., 2023), resulting in over-smoothed reconstructions where high-frequency textures are averaged out and small, low-contrast lesions are suppressed (Song et al., 2021). To recover high-frequency fidelity, the paradigm shifted towards Posterior Sampling. Early attempts using GANs (Isola et al., 2017; Xue et al., 2021; Liu et al., 2022; Manoj Doss & Chen, 2024; Wang et al., 2024) succeeded in synthesizing realistic textures but suffered from training instability and mode collapse. Recently, Score-based Generative Models (SGMs) and Diffusion Models (DDPMs) have emerged as the state-of-the-art (Ai et al., 2025; Luo et al., 2023; Gan et al., 2025; Webber et al., 2025; Yu et al., 2025; Tang et al., 2024; Shen et al., 2024). By formulating reconstruction as the time-reversal of a SDE (Song et al., 2021), these methods iteratively refine noisy priors onto the data manifold, demonstrating superior capability in modeling complex multimodal distributions.

Despite their success, these stochastic solvers fundamentally rely on *dissipative dynamics* to stabilize the generative trajectory. While effective for denoising, this contraction mechanism is indiscriminate: it aggressively suppresses low-probability features. In the low-count regime, weak lesion signals are prone to *numerical extinction*. They are effectively "cleaned away" by the dissipative flow before they can form coherent structures, leading to the "signal wash-out" phenomenon (Toth et al., 2020).

**Hamiltonian and Symplectic Dynamics** The integration of symplectic geometry into deep learning has emerged

as a powerful inductive bias for ensuring physical consistency. Building on Neural ODEs (Chen et al., 2021), Hamiltonian Neural Networks (HNNs) and Symplectic Recurrent Neural Networks were proposed to parameterize continuous-time dynamics that strictly satisfy conservation laws, such as energy conservation. Unlike unconstrained solvers, these methods leverage symplectic integrators to maintain phase-space volume (Toth et al., 2020) (Obey Liouville's Theorem (Safko et al., 2002)), proving highly effective for stable, long-horizon physical simulations (Chen et al., 2020). Beyond dynamics learning, this structure-preserving property has been extended to generative modeling. Early volume-preserving flows like NICE (Dinh et al., 2015) and Hamiltonian Generative Networks (Toth et al., 2020) exploited reversible dynamics for efficient density estimation and sampling. Recently, Symplectic Generative Networks (Aich & Aich, 2025) have further formalized this framework. However, these approaches primarily address unconditional generation, physical simulation, or spatial registration. They have yet to be effectively adapted for *ill-posed inverse problems*, where the challenge is not just conservation, but guiding the conservative flow to satisfy rigorous data consistency constraints. In this paper, **FlowPET** bridges this gap by embedding a Separable Hamiltonian structure within a conditional Flow Matching framework, leveraging geometric conservation to immunize weak PET signals against numerical dissipation.

## 3. Preliminaries

**Problem Formulation.** Low-count PET acquisition is intrinsically governed by Poisson photon-counting statistics. Given the unknown radiotracer distribution $x \in \mathbb{R}^d$ and the system matrix $A \in \mathbb{R}^{m \times d}$, the measured sinogram $y \in \mathbb{R}^m$ follows $y \sim \text{Poisson}(Ax)$. Reconstruction is thus an ill-posed inverse problem. We frame this as Bayesian inference, aiming to sample from the intractable posterior $p(x \mid y) \propto p(y \mid x)p(x)$. Due to the high dimensionality of $x$, direct sampling is computationally prohibitive. This motivates the use of continuous-time generative models to construct a transport map from a simple noise distribution to this complex target posterior.

**Conditional Flow Matching.** Flow Matching (FM) (Lipman et al., 2023; Liu et al., 2023) offers a robust, simulation-free framework for training such continuous transport maps. FM regresses a time-dependent vector field $v_t$ that generates a probability path $p_t(z)$ interpolating between a source $p_0$ and a target $p_1$. For a conditional path defined by linear interpolation $z_t = (1-t)z_0 + tz_1$, the target vector field is explicitly $u_t(z|z_1) = z_1 - z_0$. The objective is to minimize the regression loss:

$$\mathcal{L}_{\text{FM}}(\theta) = \mathbb{E}_{t, z_0, z_1} \left[ \|v_\theta(z_t, t) - (z_1 - z_0)\|^2 \right]. \quad (1)$$

Standard FM operates on Euclidean manifolds where the vector field $v_\theta$ is unconstrained. Consequently, the flow is prone to *phase-space contraction*, which can lead to the numerical extinction of weak signals in low-count regimes.

**Hamiltonian Dynamics.** To enforce volume preservation, we turn to Hamiltonian mechanics, which describes system evolution on a symplectic phase space $\mathcal{Z} = \mathcal{X} \times \mathcal{P} \cong \mathbb{R}^{2d}$. Here, the state $z = (x, p)$ augments the image data $x$ with an auxiliary momentum $p$. The dynamics are governed by a scalar Hamiltonian $H(z, t)$ via the canonical equations:

$$\frac{dz}{dt} = J\nabla_z H(z, t), \quad J = \begin{bmatrix} 0 & I_d \\ -I_d & 0 \end{bmatrix}. \quad (2)$$

A fundamental consequence of this structure is *Liouville's Theorem* (Safko et al., 2002): the vector field is divergence-free, i.e., $\nabla \cdot (J\nabla H) \equiv 0$. This guarantees that phase-space volume is strictly invariant along the trajectory (Safko et al., 2002). By parameterizing the flow via a Hamiltonian, we ensure theoretically that the probability density is transported without the artificial contraction inherent in standard Euclidean flows.

## 4. Method

**Motivation** The fundamental bottleneck in recovering low-count PET images is the fragility of weak lesion signals against the dissipative nature of standard generative models. In Euclidean image manifolds, continuous-time flows (e.g., in diffusion models) typically induce a negative divergence ($\nabla \cdot v < 0$), fundamentally relying on *phase-space contraction* to concentrate probability mass. While this contraction effectively suppresses noise, it creates a perilous environment for small, spatially compact lesions, which are prone to *numerical extinction*, effectively "washed out" alongside the noise before they can form coherent structures.

To resolve this dilemma, we argue that the generative process must be *lifted* from a dissipative geometry to a conservative one. We propose **FlowPET** (Fig. 2), which reformulates reconstruction as a trajectory in an augmented *Symplectic Phase Space*. By adopting a Hamiltonian formulation, we ensure, by construction, that the transport is strictly volume-preserving (Liouville's Theorem). This structural invariance acts as a safeguard, allowing the model to distinguish signal from noise based on rigorous symplectic flow matching rather than aggressive geometric contraction.

### 4.1. Hamiltonian Posterior Transport

Standard generative flows operate on the image manifold $\mathcal{X} \subseteq \mathbb{R}^d$. To enable non-dissipative transport, we lift the conditional reconstruction problem to the **symplectic phase space** $\mathcal{Z} := \mathcal{X} \times \mathcal{P} \cong \mathbb{R}^{2d}$. Here, $\mathcal{P} \cong \mathbb{R}^d$ serves as an auxiliary momentum buffer, allowing signal energy to be

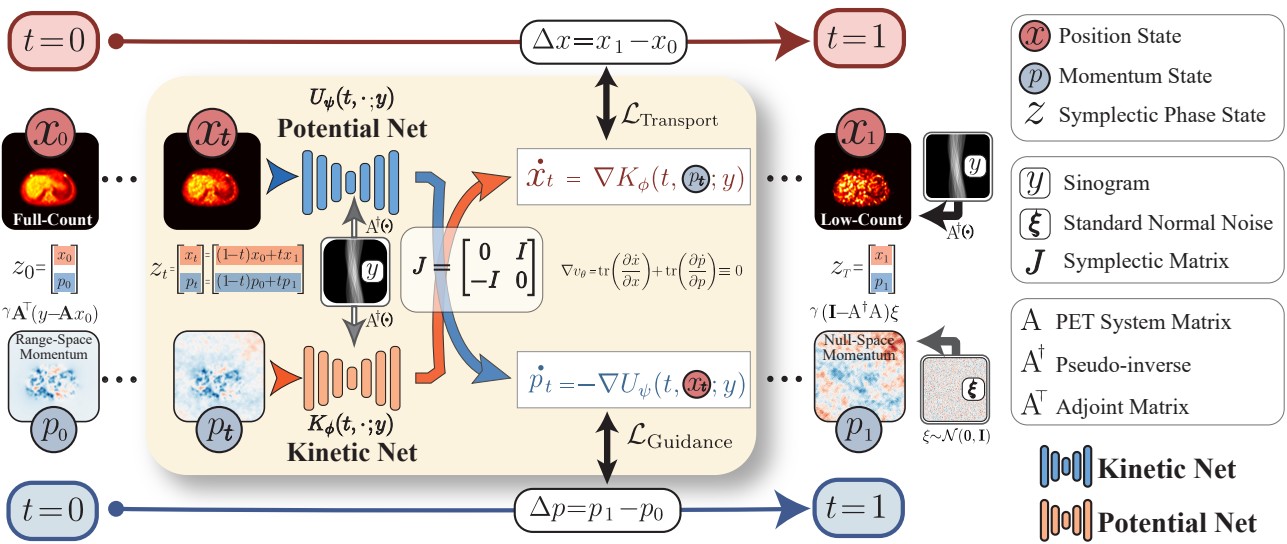

*Figure 2.* Overview of the **FlowPET** framework. **FlowPET** formulates reconstruction as volume-preserving transport in symplectic phase space $\mathcal{Z} = \mathcal{X} \times \mathcal{P}$, driven by a Separable Hamiltonian System $(U_\psi, K_\phi)$ ensuring divergence-free dynamics. The trajectory utilizes physics-informed momentum initialization, where $p_0$ embeds range-space data consistency and $p_1$ injects orthogonal null-space uncertainty to recover textures strictly adhering to physical constraints.

dynamically redistributed between structural forms (position) and latent velocities (momentum) rather than being dissipated out of the system.

For a given measurement $y$, we parameterize the posterior transport as a flow map $\phi_t(\cdot \mid y)$ governed by Hamilton's equations:

$$\dot{z} = J\nabla_z H_\theta(t, z; y), z = \begin{bmatrix} x \\ p \end{bmatrix}, J = \begin{bmatrix} 0 & I_d \\ -I_d & 0 \end{bmatrix}, \quad (3)$$

where $J$ is the canonical symplectic matrix defining the geometric structure of the flow.

**Separable Hamiltonian Parameterization.** While any Hamiltonian formulation guarantees volume preservation in continuous time, practical deployment requires an architecture that supports efficient, structure-preserving numerical integration. To this end, we impose a **Separable Hamiltonian System** inductive bias. We strictly decompose the Hamiltonian energy $H_\theta$ into a potential term $U_\psi$ (dependent only on state) and a kinetic term $K_\phi$ (dependent only on momentum):

$$H_\theta(t, x, p; y) = U_\psi(t, x; y) + K_\phi(t, p; y). \quad (4)$$

Substituting this decomposition into the symplectic gradient yields a block-structured vector field:

$$v_\theta(t, z; y) = \begin{bmatrix} \dot{x} \\ \dot{p} \end{bmatrix} = \begin{bmatrix} \nabla_p K_\phi(t, p; y) \\ -\nabla_x U_\psi(t, x; y) \end{bmatrix}. \quad (5)$$

This design ensures that the time-evolution of $x$ is driven exclusively by the momentum conjugate, while the force acting on $p$ is derived exclusively from the spatial potential.

**Proposition 4.1** (Divergence-Free by Construction). *Assume $U_\psi$ and $K_\phi$ are continuously differentiable. The Jacobian trace of the separable vector field in Eq. (5) vanishes identically due to the vanishing diagonal blocks:*

$$\nabla_z \cdot v_\theta = \text{tr}\left(\frac{\partial \dot{x}}{\partial x}\right) + \text{tr}\left(\frac{\partial \dot{p}}{\partial p}\right) = \text{tr}(\mathbf{0}) + \text{tr}(\mathbf{0}) \equiv 0. \quad (6)$$

*Proof.* See the Appendix A.2 □

**Physical Implications.** Proposition 4.1 establishes that the flow satisfies Liouville's theorem *by architectural construction*. In the context of low-count PET, this provides a topological guarantee: *the phase-space volume occupied by weak lesion signals is an invariant of the motion*. Unlike diffusion models where probability mass can contract into the mode, our Hamiltonian transport strictly conserves information density, ensuring that subtle pathologies are transported, *not erased*, during the reconstruction process.

### 4.2. Physics-Informed Symplectic Flow Modeling

While the Hamiltonian formulation (Section 4.1) guarantees volume-preserving transport, it implies a conservation law without specifying the *geometry* of the transport. In ill-posed inverse problems, the posterior landscape is fundamentally anisotropic: it is stiffly constrained in directions measurable by the forward operator, yet expansive in the null space where data is silent.

To navigate this landscape, we introduce a **Range-Null Space Decomposition** into the phase-space boundaries. We

construct conjugate boundary conditions that explicitly decouple data consistency (Range) from stochastic variability (Null), ensuring the learned flow respects the spectral properties of the PET operator.

### 4.2.1. GEOMETRIC STRUCTURE OF THE PET POSTERIOR

The system matrix $\mathbf{A}$ imposes a fundamental information asymmetry. The likelihood function constrains the solution manifold only along the range space of the adjoint $\mathbf{A}^\top$, leaving fine-scale textures, which predominantly reside in the kernel of $\mathbf{A}$, under-determined. Standard methods often conflate these subspaces, leading to a dilemma: either overfit the noise (range-space corruption) or over-smooth the texture (null-space collapse). A faithful transport mechanism must preserve orthogonality: it should strictly enforce consistency where the physics is informative, and inject controlled stochasticity only where the physics is blind.

### 4.2.2. CONJUGATE PHASE-SPACE BOUNDARIES

We define a probability path bridging the full-count reference distribution ($t = 0$) and a measurement-informed prior ($t = 1$). The forward trajectory ($t = 0 \to 1$) denotes degradation, while the reverse trajectory ($t = 1 \to 0$) performs the reconstruction. The boundary states $z_0, z_1 \in \mathcal{Z}$ are defined as:

$$z_0 = \begin{bmatrix} x_0 \\ p_0 \end{bmatrix} \quad \text{(Source)}, \qquad z_1 = \begin{bmatrix} x_1 \\ p_1 \end{bmatrix} \quad \text{(Target)}.$$

**Position Boundaries.** The spatial coordinates are anchored by the fully-sampled reference $x_0 := x_{\text{full}}$ and the pseudo-inverse approximation: $x_1 := \mathbf{A}^\dagger y$ (implemented via Filtered Back-Projection). **Momentum Boundaries.** The novelty lies in the momentum variables, which we utilize to encode the *gradient flow* of the posterior density:

**Range-Space Momentum ($t = 0$: Data Score Embedding).** At the source boundary, we define momentum to capture the driving force of the likelihood. To counteract the scale disparity induced by the ill-conditioned nature of $\mathbf{A}^\top$, we introduce the **Momentum Compression Factor** $\gamma$:

$$p_0 := \gamma \, \mathbf{A}^\top (y - \mathbf{A} x_0). \tag{7}$$

Geometrically, $p_0$ corresponds to the scaled negative gradient of an $L_2$ data fidelity surrogate $\frac{1}{2}\|y - \mathbf{A}x\|^2$, adopted in place of the exact Poisson log-likelihood to avoid the singularity of the Poisson gradient in extreme low-count regimes. By initializing $p_0$ with this value, we effectively embed the Data Score into the momentum variable. This acts as a *restoring force* within the Hamiltonian system, pulling the trajectory towards the manifold of data-consistent solutions during the generative process (See the Appendix A.3).

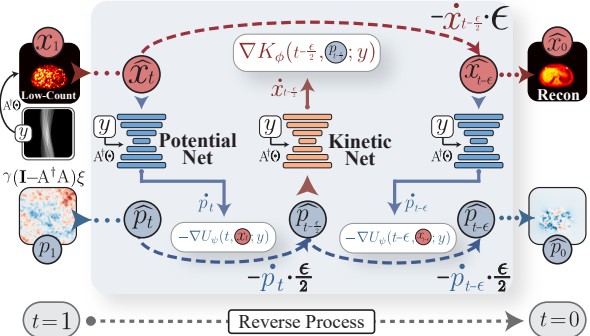

**Symplectic Leapfrog Integrator**

*Figure 3.* Symplectic Leapfrog Integrator. Leveraging the separable Hamiltonian, this time-staggered Störmer-Verlet scheme interleaves half-step momentum kicks ($\nabla U_\psi$) with full-step position drifts ($\nabla K_\phi$). This discretization maintains a unit Jacobian determinant, guaranteeing exact phase-space volume preservation.

**Null-Space Momentum ($t = 1$: Orthogonal Noise Injection).** At the target boundary, we isolate uncertainty. We initialize $p_1$ by projecting isotropic Gaussian noise onto the null space of the operator, scaled identically by $\gamma$ to ensure the stochastic energy injection is commensurate with the deterministic restoring force:

$$p_1 = \gamma \, (\mathbf{I} - \mathbf{A}^\dagger \mathbf{A})\xi, \quad \xi \sim \mathcal{N}(0, \mathbf{I}). \tag{8}$$

where $(\mathbf{I} - \mathbf{A}^\dagger \mathbf{A})$ is the orthogonal projector onto $\text{Null}(\mathbf{A})$. This construction is pivotal. Unlike standard diffusion which corrupts all components equally, this **Orthogonal Noise Injection** ensures that the generator explores diversity *only* in the unobserved subspace. It allows the model to hallucinate plausible high-frequency textures (null-space filling) without contradicting the low-frequency structural constraints (range-space backbone) imposed by the sinogram.

### 4.2.3. SYMPLECTIC FLOW MATCHING

To learn the conservative dynamics connecting these boundaries, we employ Symplectic Flow Matching (SFM). We define the time-dependent probability path via linear interpolation $z_t = (1 - t)z_0 + tz_1$, inducing a vector field $u_t = z_1 - z_0$. The generative Hamiltonian vector field $v_\theta$ is trained by minimizing the regression objective over the symplectic manifold:

$$\mathcal{L}_{\text{SFM}} = \mathbb{E}_{t, z_0, z_1} \left[ |v_\theta(t, z_t; y) - (z_1 - z_0)|^2 \right]. \tag{9}$$

Crucially, our Separable Hamiltonian parameterization (Equation (4)) allows this objective to decouple into physically interpretable components:

$$\mathcal{L}_{\text{SFM}} = \underbrace{\mathbb{E}\left[|\nabla_p K_\phi - \Delta x|^2\right]}_{\text{Kinetic Matching (Transport)}} + \underbrace{\mathbb{E}\left[|-\nabla_x U\psi - \Delta p|^2\right]}_{\text{Potential Matching (Guidance)}},$$
$$\tag{10}$$

*Table 1.* **Quantitative comparison** of reconstruction performance across BrainWeb (20% Count), In-House (1% Count), and UDPET Brain (1% Count) datasets. The best and second-best results are indicated in **bold** and underlined, respectively.

| Method | BrainWeb (20% Count) | | | In-House (1% Count) | | | UDPET Brain (1% Count) | | |
|---|---|---|---|---|---|---|---|---|---|
| | SSIM↑ | PSNR↑ | RMSE↓ | SSIM↑ | PSNR↑ | RMSE↓ | SSIM↑ | PSNR↑ | RMSE↓ |
| OSEM (Hudson & Larkin, 1994) | 0.9078 | 28.35 | 0.0447 | 0.7456 | 23.59 | 0.0745 | 0.7607 | 19.87 | 0.1108 |
| AutoContextCNN (Xiang et al., 2017) | 0.9816 | 33.64 | 0.0233 | 0.9339 | 33.66 | 0.0226 | 0.8794 | 26.29 | 0.0541 |
| DeepPET (Häggström et al., 2019) | 0.9746 | 30.08 | 0.0331 | 0.8820 | 32.24 | 0.0263 | 0.8218 | 25.28 | 0.0581 |
| CNNBPnet (Zhang et al., 2020) | 0.9560 | 30.62 | 0.0329 | 0.9240 | 34.62 | 0.0200 | 0.7750 | 25.06 | 0.0621 |
| FBPnet (Wang & Liu, 2020) | 0.9327 | 33.62 | 0.0231 | 0.9592 | 34.19 | 0.0210 | 0.8907 | 27.36 | 0.0463 |
| LCPR-Net (Xue et al., 2021) | 0.9769 | 33.75 | 0.0224 | 0.9222 | 34.95 | 0.0206 | 0.8919 | 27.77 | 0.0446 |
| Sino-cGAN (Liu et al., 2022) | 0.9641 | 30.76 | 0.0306 | 0.9704 | 33.58 | 0.0223 | 0.8646 | 25.54 | 0.0569 |
| IR-SDE (Luo et al., 2023) | 0.9589 | 32.55 | 0.0266 | 0.9728 | 33.81 | 0.0222 | 0.9103 | 25.52 | 0.0549 |
| DGLM_u (Zhang et al., 2024) | 0.9785 | 33.58 | 0.0230 | 0.9551 | 32.93 | 0.0245 | 0.8905 | 25.95 | 0.0552 |
| RED (Ai et al., 2025) | 0.9664 | 34.45 | 0.0210 | 0.9472 | 34.15 | 0.0192 | 0.8890 | 26.51 | 0.0474 |
| DREAM (Huang et al., 2025) | 0.9777 | 33.07 | 0.0246 | 0.9661 | 35.77 | 0.0173 | 0.9081 | 28.49 | 0.0411 |
| FourierPET (Zhang et al., 2026) | **0.9859** | 35.36 | 0.0198 | 0.9740 | 35.19 | 0.0188 | 0.9083 | 27.98 | 0.0437 |
| *FlowPET* (Ours) | *0.9838* | **36.34** | **0.0188** | **0.9811** | **36.35** | **0.0160** | **0.9139** | **28.88** | **0.0387** |

where $\Delta x = x_1 - x_0$ and $\Delta p = p_1 - p_0$. This decomposition simplifies optimization: $K_\phi$ learns the kinematic path of mass transport, while $U_\psi$ learns the underlying potential energy landscape that shapes the posterior geometry.

### 4.3. Structure-Preserving Discretization

The theoretical guarantees of **FlowPET**, specifically the divergence-free nature of the vector field, are derived in the continuous-time limit. However, numerical realization requires discretizing these dynamics. A critical pitfall is the use of standard solvers (e.g., Runge-Kutta (Butcher, 1996)), which are not symplectic; they introduce numerical dissipation that acts as an artificial viscosity, violating Liouville's theorem (Safko et al., 2002) and causing the phase-space volume to contract over integration steps. For low-count PET, this numerical contraction would reintroduce the very signal wash-out we aim to eliminate.

To maintain geometric consistency in the discrete domain, we employ the Symplectic Leapfrog (Störmer–Verlet (VERLET, 1968)) integrator in Fig 3. Crucially, the Separable Hamiltonian architecture (Equation (4)) we enforced earlier acts as the enabler here: it permits the use of *explicit* time-staggered updates, avoiding the prohibitive computational cost of implicit solvers required for general Hamiltonians. The update rule for a backward step $\epsilon$ (where $\epsilon > 0$ for inference) is given by:

$$\hat{p}_{t-\frac{\epsilon}{2}} = \hat{p}_t + \frac{\epsilon}{2} \nabla_x U_\psi(t, \hat{x}_t; y), \quad (11)$$

$$\hat{x}_{t-\epsilon} = \hat{x}_t - \epsilon \nabla_p K_\phi\left(t - \frac{\epsilon}{2}, \hat{p}_{t-\frac{\epsilon}{2}}; y\right), \quad (12)$$

$$\hat{p}_{t-\epsilon} = \hat{p}_{t-\frac{\epsilon}{2}} + \frac{\epsilon}{2} \nabla_x U_\psi(t - \epsilon, \hat{x}_{t-\epsilon}; y). \quad (13)$$

This discretization possesses a unique property: the Jacobian determinant of the update map is exactly unity. Consequently, phase-space volume is preserved exactly, independent of the step size $\epsilon$. Unlike standard solvers where error manifests as dissipation, the error in symplectic integration manifests as a bounded oscillation of the energy (tracking a "shadow Hamiltonian"). This provides a rigorous guarantee: the null-space uncertainty injected at the boundary is transported losslessly to the final reconstruction, immune to numerical extinction (See the proof in Appendix A.4).

**Inference Protocol.** At inference time, we sample from the posterior $p(x|y)$ by reversing the flow from the prior boundary ($t = 1$) to the data manifold ($t = 0$). We initialize the state $z_1 = (x_1, p_1)$ using the measurement-conditioned prior established in Section 4.2:

$$x_1 = \mathbf{A}^\dagger y, \quad p_1 = \gamma \left(\mathbf{I} - \mathbf{A}^\dagger \mathbf{A}\right)\xi, \quad \xi \sim \mathcal{N}(0, \mathbf{I}). \quad (14)$$

By integrating the learned dynamics backward, we recover a sample $x_0$. Because the flow is volume-preserving and the momentum initialization is orthogonal to the data, the resulting $x_0$ effectively superimposes generated null-space textures onto the range-space backbone provided by $y$, yielding a reconstruction that is both high-fidelity and physically consistent.

## 5. Experiments

### 5.1. Experimental Setup

**Datasets.** We evaluate FlowPET across three distinct low-count scenarios: (1) **BrainWeb** (Aubert-Broche et al., 2006), comprising 20 simulated volumes (3,200 slices) at 20% dose, evaluated via leave-one-out cross-validation; (2) **In-house**, containing 60 whole-body pediatric scans (40,440 slices) with synthetic 1% dose, partitioned into 48 subjects for training/validation and 12 for testing; and (3) **UDPET Brain** (Xue et al., 2022), consisting of 206 scans (26,368 slices) with a dose reduction factor (DRF) of 100, split into

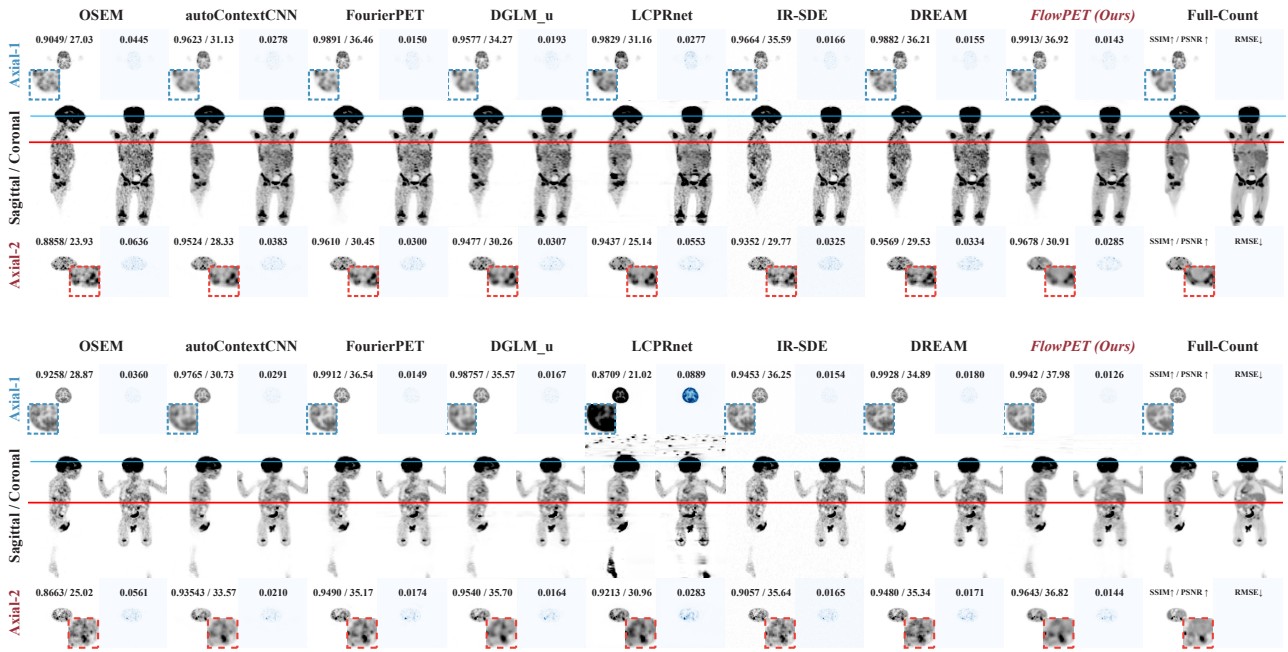

*Figure 4.* Qualitative visualization results on the clinical pediatric dataset. The figure displays Axial cross-sections of the brain (top, indicated by the blue line) and the torso (bottom, indicated by the red line) accompanied by zoomed-in patches and error maps, alongside the central Sagittal / Coronal whole-body views. Corresponding quantitative metrics above panels validate FlowPET's superior structural fidelity compared to leading baselines.

170 for training/validation and 36 for testing.

**Network Architectures.** To enforce the separable Hamiltonian structure $H(x,p) = U(x) + K(p)$, we do not parameterize the scalar energy directly. Instead, we employ two independent neural networks to approximate the conservative vector fields: the **Kinetic network** approximates the velocity $\nabla_p K_\phi(t, p; y)$, and the **Potential network** approximates the force field $-\nabla_x U_\psi(t, x; y)$. Both networks utilize a U-Net backbone adapted from Guided-Diffusion (Dhariwal & Nichol, 2021), incorporating an encoder for sinogram features $A^\top(y)$ injected via Adaptive Layer Normalization (AdaLN) (Perez et al., 2018). We adopt an asymmetric design strategy to balance expressivity and computational inference speed: the Kinetic network $K_\phi$, responsible for the primary momentum transport, features a robust capacity (128 base channels, depth multipliers $1, 2, 4, 8$). Conversely, the Potential network $U_\psi$ acts as a geometric guide and is designed as a lightweight variant (64 base channels, multipliers $1, 2, 4$), reducing the memory footprint of the dual-network system.

**Training Protocol.** The framework is implemented in PyTorch 2.1.1 on four NVIDIA RTX 3090 GPUs. We train using the AdamW optimizer ($\beta_1 = 0.9, \beta_2 = 0.999$) with a batch size of 8. The learning rate follows a cosine annealing schedule, decaying from $10^{-4}$ to $10^{-6}$ over 500k iterations. Training pairs ($128 \times 128$) are samples from the

each dataset, where full-count OSEM (Hudson & Larkin, 1994) reconstructions serve as the ground truth $x_{\text{GT}}$.

## 5.2. Comparative Analysis

**Quantitative Superiority.** As summarized in **Table 1**, **FlowPET** establishes a new state-of-the-art across all metrics. Crucially, our method successfully navigates the trade-off that limits prior arts: it surpasses deterministic methods in texture recovery (higher SSIM) without succumbing to the hallucinations or noise artifacts typical of stochastic solvers (lower RMSE). We provide a detailed analysis of these performance gains in **Appendix C**.

**Qualitative Superiority.** Visual comparisons on the pediatric dataset (Figure 4) reveal the "signal wash-out" phenomenon inherent in dissipative baselines. As highlighted in the zoomed-in regions, **FlowPET** treats the weak lesion signal as an invariant mass to be transported rather than noise to be dissipated, yielding structural fidelity closest to the full-count reference. Additional visual comparisons and a pixel-level error map analysis are provided in **Appendix C**.

## 5.3. Ablation Studies and Analysis

**Ablation on Range-Space and Null-Space Momentum.** To validate our physics-informed phase space design, we ablate the Range-Space (**Restoring**, $p_0$) and Null-Space (**Thermal**, $p_1$) momentum components. As detailed in Ta-

*Table 2.* Ablation of Range-Space (**Restoring**, $p_0$) and Null-Space (**Thermal**, $p_1$) momentum. Best results are **bolded**.

| Restoring ($p_0$) | Thermal ($p_1$) | In-House Whole-Body | | | UDPET Brain | | |
|---|---|---|---|---|---|---|---|
| | | SSIM↑ | PSNR↑ | RMSE↓ | SSIM↑ | PSNR↑ | RMSE↓ |
| | | 0.9711 | 35.96 | 0.0166 | 0.8915 | 28.64 | 0.0398 |
| ✓ | | 0.9766 | **36.36** | **0.0159** | 0.9047 | 28.50 | 0.0409 |
| | ✓ | 0.9774 | 35.87 | 0.0169 | 0.9021 | 28.63 | 0.0401 |
| ✓ | ✓ | **0.9811** | 36.35 | 0.0160 | **0.9139** | **28.88** | **0.0387** |

*Table 3.* Comparison of Momentum Compression Factor $\gamma$ of momentum on the **In-House Whole-Body Scans**. Best results are **bolded**.

| Parameter $\gamma$ | SSIM ↑ | PSNR ↑ | RMSE ↓ |
|---|---|---|---|
| 1 | 0.9616 | 35.67 | 0.0172 |
| $10^{-1}$ | 0.9749 | 36.15 | 0.0163 |
| $10^{-2}$ | **0.9811** | **36.35** | **0.0160** |
| $10^{-3}$ | 0.9796 | 35.98 | 0.0167 |

ble 2, the baseline configuration yields the lowest metrics, confirming that unguided momentum initialization is insufficient. Incorporating the Restoring force significantly enhances fidelity, increasing Brain SSIM by 0.0132 over the baseline; this confirms the necessity of embedding data consistency gradients. Crucially, the complete **FlowPET** framework, which integrates the orthogonal Thermal component, achieves the best structural recovery. It attains the highest SSIM on both Whole-Body and UDPET Brain datasets. While the deterministic Restoring-only configuration yields marginally higher PSNR, the combined model secures superior structural similarity, demonstrating that decoupling deterministic consistency from stochastic uncertainty is essential for capturing fine-grained anatomical details.

**Sensitivity to Momentum Compression Factor $\gamma$.** Table 3 investigates the reconstruction sensitivity on Whole-Body scans across logarithmic scales, and we observe a distinct performance peak at $\gamma = 10^{-2}$. A large $\gamma$ ($\gamma \geq 10^{-1}$) amplifies the gradients from $\mathbf{A}^{\top}$, injecting excessive kinetic energy. This results in "stiff" dynamics where the restoring force overshoots the manifold, leading to optimization instability and reduced SSIM. An overly small $\gamma$ ($\gamma \leq 10^{-3}$) dampens the momentum to near-zero. This dilutes the physical guidance ($p_0$) and limits the null-space exploration ($p_1$), causing the model to degenerate into a standard position-only regression. Consequently, we adopt $\gamma = 10^{-2}$ as the canonical setting to ensure optimal transport stability.

**Additional further studies are provided in Appendix B.**

### 5.4. Wash-Out Analysis

To rigorously quantify the signal extinction phenomenon, we employ a **Signal Recovery Trajectory Analysis** on two synthetic benchmarks designed to isolate preservation capabilities: the **Spherical Lesion Phantom** (point sources, varying FWHM/contrast) and the **Synthetic Le-**

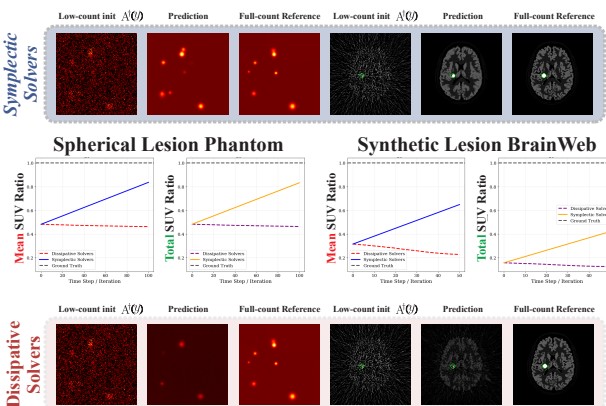

*Figure 5.* Quantitative analysis of the signal wash-out phenomenon on Spherical Lesion Phantom and Synthetic Lesion BrainWeb datasets. The curves track the evolution of Mean and Total SUV Ratios during the reverse reconstruction process.

**sion BrainWeb** (anatomically complex background). Distinct from standard endpoint evaluations, we monitor the evolution of the signal $\hat{x}_t$ along the discrete reverse trajectory ($t = 1 \rightarrow 0$, 100 steps). At each step, we compute both the **Mean** and **Total Relative SUV Ratios**, defined as the ratio of the recovered Standard Uptake Value (SUV) statistics within the lesion Region of Interest (ROI) to the ground truth. Crucially, this metric serves as a deterministic proxy for probability mass conservation: a ratio approaching 1.0 indicates that the generative flow has successfully transported the lesion's signal mass from the null space to the image space, whereas a drop indicates "wash-out" or mass leakage into the background.

As illustrated in Figure 5, the trajectories reveal a fundamental dynamic divergence. The **Dissipative Solver** (Luo et al., 2023) (dashed lines) exhibits characteristic energy collapse: the SUV ratios rise initially but suffer from *premature saturation*, plateauing well below the ground truth. This confirms that the dissipative contraction aggressively suppresses weak signals, treating them as noise to be smoothed. In contrast, **FlowPET** (solid lines) demonstrates a sustained, monotonic recovery towards the baseline (1.0). By strictly enforcing symplectic conservation, our integrator shields the fragile lesion signal from numerical extinction, allowing the data-driven restoring force to fully reconstruct the pathological intensity.

## 6. Conclusion

In this work, we presented **FlowPET** to resolve the signal "wash-out" pathology in low-count PET by replacing dissipative diffusion with volume-preserving symplectic transport. By embedding the Range-Null space decomposition into the Separable Hamiltonian dynamics, our framework successfully decouples the restoration of measurement-consistent structures from the stochastic exploration of texture. This

design effectively immunizes diagnostically vital lesions against phase-space contraction while maintaining high structural fidelity. Our results establish symplectic geometry not merely as a tool for physical simulation, but as a robust geometric safeguard for solving ill-posed inverse problems where information conservation is as critical as noise removal. Future work will explore extending this conservative paradigm to dynamic PET imaging and other scientific domains governed by strict conservation laws.

## Acknowledgments

This work was supported by a Shenzhen-Hong Kong-Macao Science and Technology Plan Project (Category C Project) under the Shenzhen Municipal Science and Technology Innovation Commission (Project No. SGDX20230821092359002) and the Collaborative Research with World-leading Research Groups scheme of The Hong Kong Polytechnic University (Project No. G-SACF).

## Impact Statement

This paper presents work whose goal is to advance the field of machine learning. There are many potential societal consequences of our work, none of which we feel must be specifically highlighted here.

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

# A. Algorithm and Proofs

## A.1. FlowPET Training and Inference Algorithm

---

**Algorithm 1 Training**: Symplectic Flow Matching

---

**Require:** Dataset $\mathcal{D} = \{(x_{\text{Full}}, y)\}$, System Matrix A
**Require:** Potential Net $U_\psi$, Kinetic Net $K_\phi$
1: **Hyperparameters:** $\gamma$ (guidance weight)
2: **repeat**
3:   Sample batch $(x_0, y) \sim \mathcal{D}$ where $x_0 = x_{\text{Full}}$
4:   *# 1. Phase-Space Boundaries (Range-Null)*
5:   $x_1 \leftarrow \text{A}^\dagger y$ {FBP Prior}
6:   $p_0 \leftarrow \text{A}^\top (y - \text{A}x_0)$ {Range-Space Momentum}
7:   $\xi \sim \mathcal{N}(0, I)$
8:   $p_1 \leftarrow (I - \text{A}^\dagger \text{A})\xi$ {Null-Space Momentum}
9:   *# 2. Optimal Transport Interpolation*
10:   $t \sim \mathcal{U}[0, 1]$
11:   $x_t \leftarrow (1 - t)x_0 + tx_1$
12:   $p_t \leftarrow (1 - t)p_0 + tp_1$
13:   *# 3. Vector Field Prediction*
14:   $v_t \leftarrow (x_1 - x_0, p_1 - p_0)$ {Target Flow}
15:   $\hat{v}_\theta \leftarrow (\nabla_p K_\phi(t, p_t; y), -\nabla_x U_\psi(t, x_t); y)$
16:   *# 4. Optimization*
17:   $\mathcal{L}_{\text{kinetic}} \leftarrow \|\nabla_p K_\phi - (x_1 - x_0)\|^2$
18:   $\mathcal{L}_{\text{potential}} \leftarrow \| - \nabla_x U_\psi - \gamma (p_1 - p_0)\|^2$
19:   $\mathcal{L} \leftarrow \mathcal{L}_{\text{kinetic}} + \mathcal{L}_{\text{potential}}$
20:   Update $\phi, \psi$ using $\nabla \mathcal{L}$
21: **until** converged

---

**Algorithm 2 Inference**: Symplectic Leapfrog

---

**Require:** Measurement $y$, Matrix $\mathbf{A}$
**Require:** Trained Nets $U_\psi, K_\phi$, Steps $N$
**Ensure:** Recon Image $x_{\text{recon}}$
1:   *# 1. Initialize at t=1 (Prior, Eq. 14)*
2:   $x_1 \leftarrow \mathbf{A}^\dagger y$
3:   $\xi \sim \mathcal{N}(0, \mathbf{I})$
4:   $p_1 \leftarrow (\mathbf{I} - \mathbf{A}^\dagger \mathbf{A})\xi$
5:   $\epsilon \leftarrow 1/N$   {Step Size, $\epsilon > 0$}
6:   $x_{\text{curr}} \leftarrow x_1, \quad p_{\text{curr}} \leftarrow p_1$
7:   *# 2. Backward Integration (t: 1 → 0)*
8:   **for** $i = N$ **down to** 1 **do**
9:     $t \leftarrow i/N$   {Current Time}
10:     *# A. Half-Step Kick (Eq. 11)*
11:     $g_{\text{pot}} \leftarrow -\nabla_x U_\psi(t, x_{\text{curr}}; y)$
12:     $p_{\text{half}} \leftarrow p_{\text{curr}} - \frac{\epsilon}{2} \cdot g_{\text{pot}}$
13:     *# B. Full-Step Drift (Eq. 12)*
14:     $t_{\text{mid}} \leftarrow t - \epsilon/2$
15:     $g_{\text{kin}} \leftarrow \nabla_p K_\phi(t_{\text{mid}}, p_{\text{half}}; y)$
16:     $x_{\text{next}} \leftarrow x_{\text{curr}} - \epsilon \cdot g_{\text{kin}}$
17:     *# C. Half-Step Kick (Eq. 13)*
18:     $t_{\text{next}} \leftarrow t - \epsilon$
19:     $g'_{\text{pot}} \leftarrow -\nabla_x U_\psi(t_{\text{next}}, x_{\text{next}}; y)$
20:     $p_{\text{next}} \leftarrow p_{\text{half}} - \frac{\epsilon}{2} \cdot g'_{\text{pot}}$
21:     $x_{\text{curr}} \leftarrow x_{\text{next}}$
22:     $p_{\text{curr}} \leftarrow p_{\text{next}}$
23:   **end for**
24:   **return** $x_{\text{recon}} \leftarrow x_{\text{curr}}$

---

## A.2. Proof of Proposition 4.1

**Proposition A.1** (Restatement of Proposition 4.1). *Assume the potential term $U_\psi(t, x; y)$ and the kinetic term $K_\phi(t, p; y)$ are continuously differentiable with respect to the state $x \in \mathcal{X}$ and momentum $p \in \mathcal{P}$, respectively. The vector field $v_\theta(t, z; y)$ induced by the Separable Hamiltonian System defined in Eq. (4) is divergence-free, satisfying:*

$$\nabla_z \cdot v_\theta(t, z; y) \equiv 0, \quad \forall z \in \mathcal{Z}. \tag{15}$$

*Proof.* The proof proceeds by explicitly deriving the Jacobian of the vector field and analyzing the trace of its block components.

**Step 1: Formulation of the Symplectic Vector Field.**  Recall that the phase-space state is defined as $z = [x^\top, p^\top]^\top \in \mathbb{R}^{2d}$ in Section 4. The dynamics are governed by the canonical equations of motion derived from the separable Hamiltonian $H_\theta(t, x, p) = U_\psi(t, x) + K_\phi(t, p)$. The induced vector field $v_\theta$ is given by the symplectic gradient:

$$v_\theta(t, z; y) = \begin{bmatrix} \dot{x} \\ \dot{p} \end{bmatrix} = J \nabla_z H_\theta = \begin{bmatrix} \mathbf{0}_d & \mathbf{I}_d \\ -\mathbf{I}_d & \mathbf{0}_d \end{bmatrix} \begin{bmatrix} \nabla_x U_\psi \\ \nabla_p K_\phi \end{bmatrix} = \begin{bmatrix} \nabla_p K_\phi(t, p; y) \\ -\nabla_x U_\psi(t, x; y) \end{bmatrix}, \tag{16}$$

where $J$ denotes the standard symplectic matrix.

**Step 2: Analysis of the Jacobian Matrix.** The divergence of the vector field corresponds to the trace of its Jacobian matrix $J_{v_\theta} = \frac{\partial v_\theta}{\partial z} \in \mathbb{R}^{2d \times 2d}$. By differentiating Eq. (16) with respect to $z$, we obtain the block matrix structure:

$$J_{v_\theta} = \begin{bmatrix} \frac{\partial \dot{x}}{\partial x} & \frac{\partial \dot{x}}{\partial p} \\ \frac{\partial \dot{p}}{\partial x} & \frac{\partial \dot{p}}{\partial p} \end{bmatrix} = \begin{bmatrix} \frac{\partial}{\partial x}(\nabla_p K_\phi) & \frac{\partial}{\partial p}(\nabla_p K_\phi) \\ \frac{\partial}{\partial x}(-\nabla_x U_\psi) & \frac{\partial}{\partial p}(-\nabla_x U_\psi) \end{bmatrix}. \tag{17}$$

**Step 3: Vanishing Diagonal Blocks via Separability.** We exploit the structural constraints imposed by the Separable Hamiltonian parameterization:

1. The kinetic term $K_\phi$ is a function solely of momentum $p$. Thus, its gradient $\nabla_p K_\phi$ is independent of position $x$, implying the upper-left diagonal block vanishes:

$$\frac{\partial}{\partial x}(\nabla_p K_\phi) = \mathbf{0}_{d \times d}. \tag{18}$$

2. Similarly, the potential term $U_\psi$ is a function solely of position $x$. Thus, its gradient $-\nabla_x U_\psi$ is independent of momentum $p$, implying the lower-right diagonal block vanishes:

$$\frac{\partial}{\partial p}(-\nabla_x U_\psi) = \mathbf{0}_{d \times d}. \tag{19}$$

Substituting these results back into the Jacobian yields a block-diagonal matrix with zero diagonals:

$$J_{v_\theta} = \begin{bmatrix} \mathbf{0}_{d \times d} & \nabla^2_{pp} K_\phi \\ -\nabla^2_{xx} U_\psi & \mathbf{0}_{d \times d} \end{bmatrix}. \tag{20}$$

**Conclusion.** The divergence is defined as the sum of the diagonal elements of the Jacobian. Therefore:

$$\nabla_z \cdot v_\theta = \text{tr}(J_{v_\theta}) = \text{tr}(\mathbf{0}_{d \times d}) + \text{tr}(\mathbf{0}_{d \times d}) = 0. \tag{21}$$

This confirms that the generated flow preserves phase-space volume element $dx \wedge dp$ by construction (Liouville's Theorem (Safko et al., 2002)). $\quad\square$

### A.3. Physical Interpretation of Range-Space Momentum

Here we justify the initialization of the Range-Space momentum $p_0$ (Eq. 7) as embedding the magnitude-scaled data score.

*Proof.* We adopt a Gaussian approximation to the Poisson log-likelihood, yielding the L2 data fidelity potential:

$$\mathcal{E}_{data}(x) = \frac{1}{2}\|y - Ax\|^2, \tag{22}$$

where $A$ is the system matrix. The "physical force" exerted by this potential on the image state $x$ is given by the negative gradient:

$$F_{data}(x) = -\nabla_x \mathcal{E}_{data}(x) \tag{23}$$

$$= -\nabla_x \left( \frac{1}{2}(y - Ax)^\top (y - Ax) \right) \tag{24}$$

$$= A^\top (y - Ax). \tag{25}$$

In our framework, we define the source momentum $p_0$ incorporating the Momentum Compression Factor $\gamma$:

$$p_0 := \gamma A^\top (y - Ax_0). \tag{26}$$

Therefore, $p_0$ corresponds to the negative gradient of the potential, **rescaled** by the preconditioning factor $\gamma$:

$$p_0 = -\gamma \nabla_x \mathcal{E}_{data}(x)|_{x=x_0}. \tag{27}$$

**Physical Implication:** By initializing the momentum with the scaled negative gradient, we provide an initial velocity that pushes the trajectory towards the low-energy manifold (where $Ax \approx y$). Crucially, the scalar $\gamma$ acts as a *linear time-invariant preconditioner*. It preserves the direction of the *restoring force* (ensuring data consistency guidance) while normalizing its magnitude to match the dynamic range of the symplectic transport, thereby preventing scale disparity caused by the ill-conditioned operator $A^\top$. $\quad\square$

## A.4. Volume Preservation in Symplectic Leapfrog Integration

We prove that the Symplectic Leapfrog integrator, employed for the reverse diffusion process, strictly preserves the phase-space volume at every discretization step. This ensures that the continuous-time guarantee of Liouville's Theorem (Proposition 4.1) translates effectively to the numerical domain, preventing the "signal wash-out" phenomenon described in the main text.

**Theorem A.2** (Discrete Liouville Property). *Let $\Phi_\epsilon : \mathcal{Z} \rightarrow \mathcal{Z}$ denote the discrete mapping defined by one step of the Symplectic Leapfrog integrator with step size $\epsilon$, mapping the state $\hat{z}_t = (\hat{x}_t, \hat{p}_t)$ to $\hat{z}_{t-\epsilon} = (\hat{x}_{t-\epsilon}, \hat{p}_{t-\epsilon})$. The Jacobian determinant of this mapping is unity:*

$$|\det(\mathbf{J}_{\Phi_\epsilon})| = \left|\det \frac{\partial(\hat{x}_{t-\epsilon}, \hat{p}_{t-\epsilon})}{\partial(\hat{x}_t, \hat{p}_t)}\right| \equiv 1. \tag{28}$$

*Proof.* Based on Eqs. 11-13 in the main text and Algorithm 2, the transition $\hat{z}_t \rightarrow \hat{z}_{t-\epsilon}$ is a composition of three sub-transformations: $\Phi_\epsilon = T_3 \circ T_2 \circ T_1$. We analyze the Jacobian of each sub-step individually.

**Step 1: First Momentum Kick** ($T_1$). The first half-step update for momentum is given by:

$$\hat{x}^* = \hat{x}_t, \tag{29}$$

$$\hat{p}^* = \hat{p}_t + \frac{\epsilon}{2}\nabla_x U_\psi(t, \hat{x}_t). \tag{30}$$

The Jacobian matrix $\mathbf{J}_1 = \frac{\partial(\hat{x}^*, \hat{p}^*)}{\partial(\hat{x}_t, \hat{p}_t)}$ has a block lower-triangular structure:

$$\mathbf{J}_1 = \begin{bmatrix} \mathbf{I}_d & \mathbf{0} \\ \frac{\epsilon}{2}\nabla_x^2 U_\psi(t, \hat{x}_t) & \mathbf{I}_d \end{bmatrix}, \tag{31}$$

where $\nabla_x^2 U_\psi$ is the Hessian of the potential. The determinant is the product of the diagonal blocks: $\det(\mathbf{J}_1) = \det(\mathbf{I}_d) \cdot \det(\mathbf{I}_d) = 1$.

**Step 2: Position Drift** ($T_2$). The full-step position update is defined as:

$$\hat{x}^{**} = \hat{x}^* - \epsilon\nabla_p K_\phi(t - \frac{\epsilon}{2}, \hat{p}^*), \tag{32}$$

$$\hat{p}^{**} = \hat{p}^*. \tag{33}$$

The Jacobian $\mathbf{J}_2$ exhibits a block upper-triangular structure, as $\hat{x}^{**}$ depends on $\hat{p}^*$ but $\hat{p}^{**}$ is merely a copy of $\hat{p}^*$:

$$\mathbf{J}_2 = \begin{bmatrix} \mathbf{I}_d & -\epsilon\nabla_p^2 K_\phi(t - \frac{\epsilon}{2}, \hat{p}^*) \\ \mathbf{0} & \mathbf{I}_d \end{bmatrix}. \tag{34}$$

Similarly, $\det(\mathbf{J}_2) = 1$.

**Step 3: Second Momentum Kick** ($T_3$). The final half-step momentum update is:

$$\hat{x}_{t-\epsilon} = \hat{x}^{**}, \tag{35}$$

$$\hat{p}_{t-\epsilon} = \hat{p}^{**} + \frac{\epsilon}{2}\nabla_x U_\psi(t - \epsilon, \hat{x}^{**}). \tag{36}$$

This mirrors the structure of $T_1$, yielding a Jacobian $\mathbf{J}_3$ with $\det(\mathbf{J}_3) = 1$.

**Conclusion.** By the chain rule, the determinant of the full step is the product of the determinants of the sub-steps:

$$\det(\mathbf{J}_{\Phi_\epsilon}) = \det(\mathbf{J}_3) \cdot \det(\mathbf{J}_2) \cdot \det(\mathbf{J}_1) = 1 \cdot 1 \cdot 1 = 1. \tag{37}$$

This result holds theoretically regardless of the step size $\epsilon$ or the complexity of the neural networks $U_\psi$ and $K_\phi$, confirming that the numerical integration is exactly volume-preserving (symplectic). □

*Table 4.* Comparison of solvers on the **Synthetic Lesion BrainWeb** from 5.4. Best results are **bolded**.

| Solvers | SSIM ↑ | PSNR ↑ | Lesion Contrast ↑ |
|---|---|---|---|
| Euler | 0.7401 | 22.89 | 0.8083 |
| RK4 (Butcher, 1996) | 0.7459 | 23.50 | 0.8244 |
| Leapfrog | **0.7460** | **23.52** | **0.8267** |

# B. Further Analyses

### B.1. Analysis of Integrator Numerical Dissipation

To validate the critical role of geometric conservation, we conducted an ablation study on the Integrator Numerical Dissipation. We fixed the trained Potential ($U_\psi$) and Kinetic ($K_\phi$) networks and performed inference on 20 Synthetic Lesion BrainWeb cases from 5.4 ($N = 100$ steps) using three solvers: Euler, Runge–Kutta (RK4) (Butcher, 1996), and our Symplectic Leapfrog.

The results in Table 4 confirm that non-symplectic solvers introduce artificial viscosity. While the high-order RK4 yields competitive structural metrics (SSIM 0.7459) compared to Euler (0.7401), Symplectic Leapfrog achieves the highest Lesion Contrast (0.8267), surpassing both RK4 (0.8244) and Euler (0.8083). This evidenced that the superior recovery of weak signals is not merely due to network capacity, but strictly stems from the Leapfrog integrator's Volume-Preserving property, which prevents the numerical "wash-out" of diagnostic features observed in dissipative solvers.

### B.2. Impact of Range-Null Momentum Decomposition

*Table 5.* Ablation of phase-space boundary strategies on UDPET Brain. We compare **Unguided** ($p_0 = 0$), **Isotropic** ($p_1 \sim \mathcal{N}$), and **Orthogonal** (Ours) strategies. Best results are **bolded**.

| Restoring Force ($p_0$) | Thermal Noise ($p_1$) | SSIM↑ | PSNR↑ | RMSE↓ |
|---|---|---|---|---|
| None (Zeros) | Null-Space Noise* | 0.9021 | **28.95** | **0.0383** |
| Data Score* | Isotropic Gaussian | 0.9112 | 28.79 | 0.0387 |
| **Data Score*** | **Null-Space Noise*** | **0.9139** | 28.88 | 0.0387 |

To isolate the contributions of our Range-Null decomposition, we compare three momentum initialization strategies (Table 5):(1) **Unguided Baseline:** $p_0 = 0$ implies no restoring force;(2) **Naive Guidance:** $p_0$ uses the data score, but $p_1$ injects isotropic Gaussian noise, ignoring operator orthogonality;(3) **FlowPET (Ours):** Fully decoupled physics-informed boundaries.

**Results Analysis.** The comparison reveals two critical insights:

- **Necessity of Restoring Force ($p_0$):** Comparing Row 1 and Row 3, introducing the Data Score significantly boosts SSIM (+0.0118). While the unguided baseline achieves the highest PSNR, this is a classic signature of *regression-to-the-mean*: without a restoring force to pull the trajectory onto the sharp data manifold, the model defaults to a low-variance, over-smoothed average.

- **Superiority of Orthogonal Injection ($p_1$):** Comparing Row 2 and Row 3 validates the Range-Null hypothesis. Injecting isotropic noise (Row 2) degrades structural fidelity (lower SSIM) because it corrupts the valid *Range Space* information. By confining uncertainty strictly to the *Null Space* (Row 3), FlowPET synthesizes texture without conflicting with the measurement consistency, achieving the optimal balance of structural fidelity and perceptual realism.

# C. Additional Analysis and Evaluation

**Comprehensive Quantitative Analysis**    As detailed in Table 1, **FlowPET** establishes a new state-of-the-art benchmark, exhibiting particularly robust generalization in severe low-count scenarios. While competitive on the standard BrainWeb benchmark (20% count), where it achieves top-tier fidelity comparable to the specialized FourierPET (Zhang et al., 2026), its advantage becomes decisive in the high-noise asymptotic regimes of the In-House and UDPET Brain datasets (1% count).

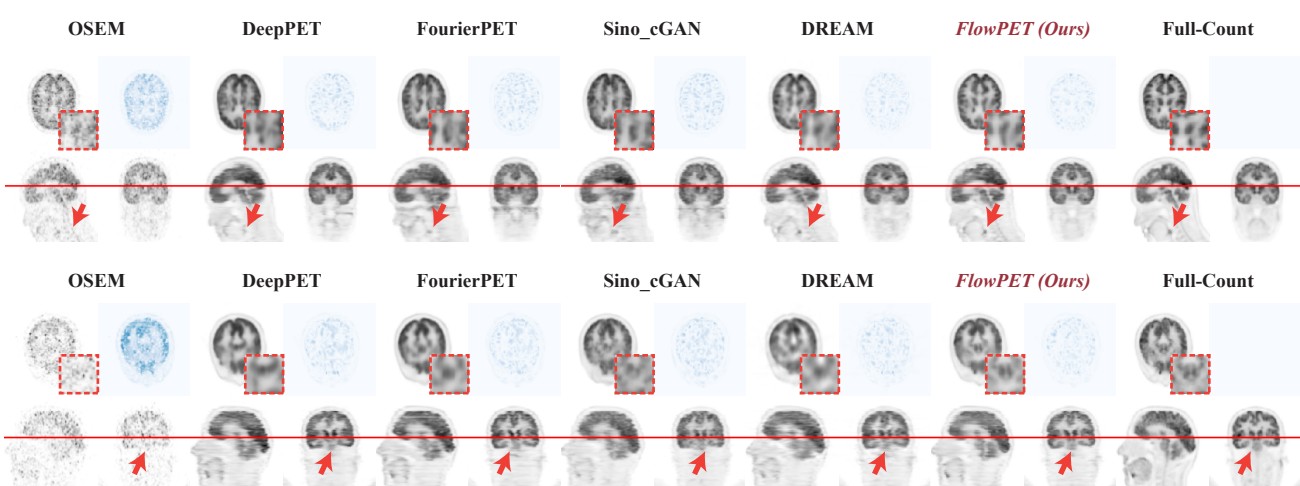

*Figure 6.* Qualitative visualization results on the UDPET Brain (1% Count) dataset. The figure displays Axial accompanied by zoomed-in patches and error maps, alongside the central Sagittal / Coronal views.

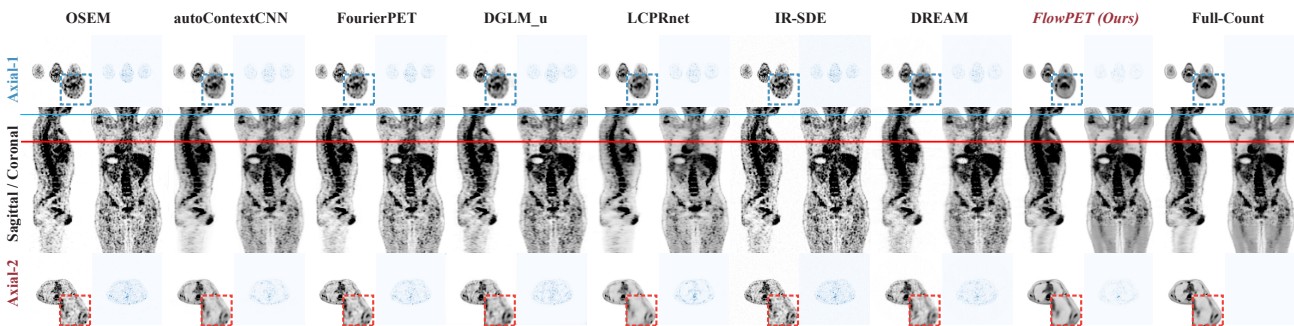

*Figure 7.* Qualitative visualization results on the clinical pediatric dataset. The figure displays Axial cross-sections of the brain (top, indicated by the blue line) and the torso (bottom, indicated by the red line) accompanied by zoomed-in patches and error maps, alongside the central Sagittal / Coronal whole-body views.

Specifically, on the challenging In-House dataset, FlowPET attains a PSNR of 36.35 **dB** and an SSIM of 0.9811, surpassing the leading stochastic generative baseline, DREAM (Huang et al., 2025), by a substantial margin of 0.58 dB. This empirical evidence confirms that symplectic transport offers a superior geometric prior for navigating the extreme ill-posedness of $100\times$ dose reduction tasks.

**Mechanism of Superiority.** FlowPET resolves the trade-off plaguing prior arts:

- **vs. Deterministic Regression:** Unlike methods prone to regression-to-the-mean (e.g., AutoContextCNN (Xiang et al., 2017), DGLM_u (Zhang et al., 2024) and FourierPET (Zhang et al., 2026)), FlowPET utilizes orthogonal null-space injection to sample the full posterior, preserving high-frequency texture that regression averages out.

- **vs. Dissipative Generation:** Contrary to standard diffusion models (e.g., IR-SDE (Luo et al., 2023) and DREAM (Huang et al., 2025)) where dissipative dynamics inadvertently suppress weak signals via phase-space contraction, FlowPET enforces **volume-preserving symplectic transport**. This geometric safeguard treats faint pathological signals as invariant mass to be transported rather than noise to be dissipated, ensuring superior recovery of low-contrast lesions without numerical extinction.

**Extended Qualitative Assessment** Figures 6 to 12 present more visual results of the compared methods, further illustrating the reconstruction performance of FlowPET relative to competing baselines across diverse anatomical sections.

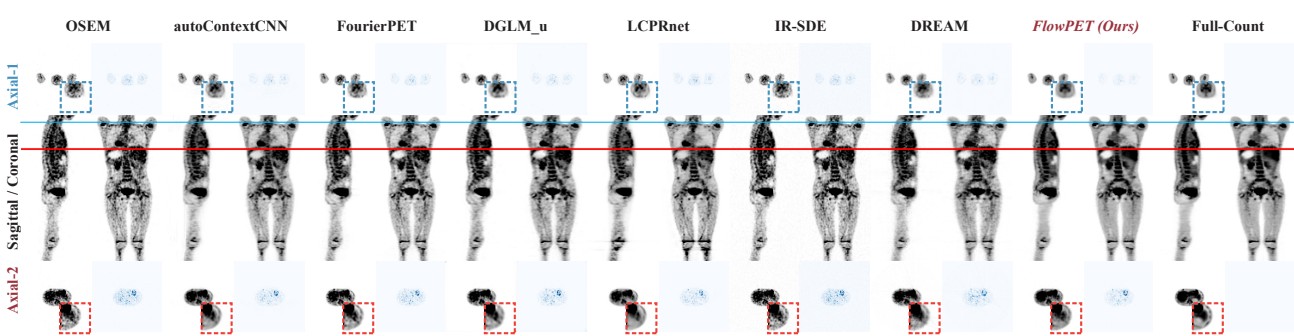

*Figure 8.* Qualitative visualization results on the clinical pediatric dataset. The figure displays Axial cross-sections of the brain (top, indicated by the blue line) and the torso (bottom, indicated by the red line) accompanied by zoomed-in patches and error maps, alongside the central Sagittal / Coronal whole-body views.

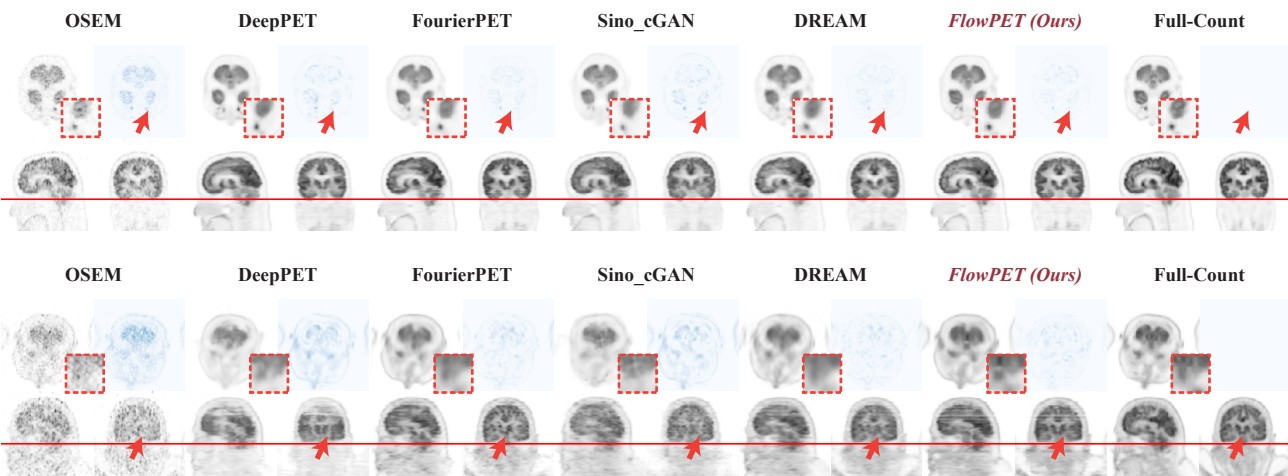

*Figure 9.* Qualitative visualization results on the UDPET Brain (1% Count) dataset. The figure displays Axial accompanied by zoomed-in patches and error maps, alongside the central Sagittal / Coronal views.

## D. Limitations and Future Work

**Limitations**  A critical constraint is that, unlike dissipative solvers that naturally contract noise, the proposed volume-preserving dynamics lack an intrinsic mechanism to stabilize convergence. This risks phase-space oscillations if the learned potential energy surface is imperfect. Furthermore, the computational overhead of the iterative Symplectic Leapfrog integrator remains a significant bottleneck for real-time clinical deployment compared to single-step methods.

**Future Work**  Research must prioritize rigorous Out-of-Distribution (OOD) generalization, validating robustness across diverse scanner geometries and acquisition protocols to ensure the model does not overfit to specific physical constraints. Additionally, the evaluation landscape should expand beyond pixel-level metrics to include quantitative clinical assessments, such as observer studies for lesion detectability, to confirm the method's diagnostic utility in practice.

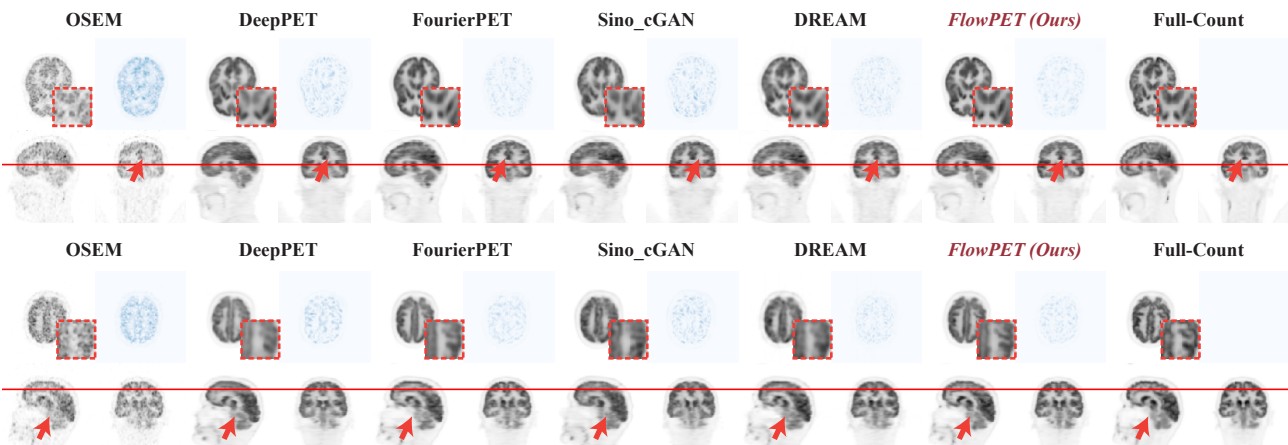

*Figure 10.* Qualitative visualization results on the UDPET Brain (1% Count) dataset. The figure displays Axial accompanied by zoomed-in patches and error maps, alongside the central Sagittal / Coronal views.

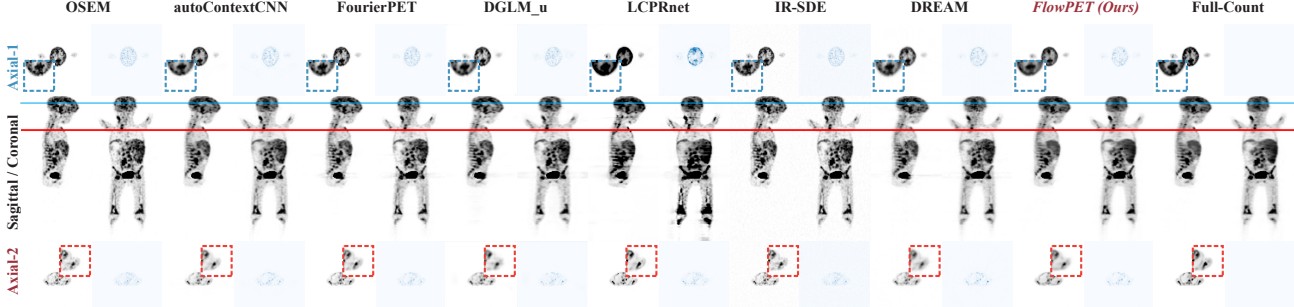

*Figure 11.* Qualitative visualization results on the clinical pediatric dataset. The figure displays Axial cross-sections of the brain (top, indicated by the blue line) and the torso (bottom, indicated by the red line) accompanied by zoomed-in patches and error maps, alongside the central Sagittal / Coronal whole-body views.

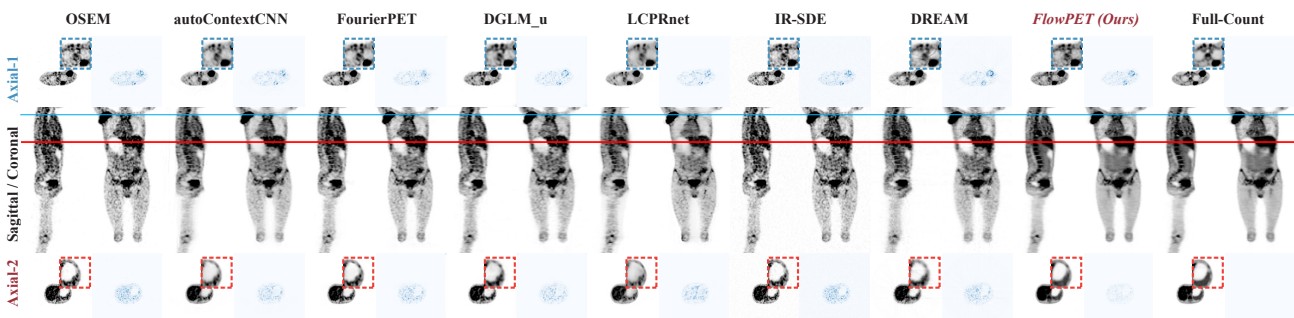

*Figure 12.* Qualitative visualization results on the clinical pediatric dataset. The figure displays Axial cross-sections of the brain (top, indicated by the blue line) and the torso (bottom, indicated by the red line) accompanied by zoomed-in patches and error maps, alongside the central Sagittal / Coronal whole-body views.

