# OpenReview forum: "FlowPET: Physics-Informed Symplectic Flow Matching for Low-Count PET Reconstruction"
_ICML.cc/2026/Conference — ICML 2026 regular_

### Official Review · Reviewer_LjC4 · 2026-02-14

**Soundness:** 2
**Presentation:** 3
**Significance:** 2
**Originality:** 3
**Overall Recommendation:** 3
**Confidence:** 2

**Summary:**

This paper introduces FlowPET, a physics-informed generative framework designed for low-count Positron Emission Tomography (PET) reconstruction. The work primarily addresses the issue of lesion signal "wash-out" or numerical extinction. Specifically, the method lifts the reconstruction problem into a Symplectic Phase Space by augmenting the image data with auxiliary momentum variables and parameterizes the posterior transport using Separable Hamiltonian Dynamics. To guide this conservative flow toward physically consistent solutions, the framework employs a Range-Null space decomposition for the boundary conditions: data consistency is encoded into the momentum variable (as a restoring force), while stochastic uncertainty is confined to the null space (as a thermal force). The authors provide experimental evidence to substantiate their claims.

**Compliance With Llm Reviewing Policy:**

Affirmed.

**Key Questions For Authors:**

Overall, the proposed method appears technically sound. However, I have two primary concerns:

Computational Cost: I am concerned about the computational overhead introduced by the proposed method. Could the authors clarify the specific increase in inference time and memory usage compared to the baseline models?

Novelty of Range-Null Space Decomposition: The Range-Null Space Decomposition [1] technique is already well-established. Consequently, the overall framework appears to be a combination of existing pipelines. I would be willing to raise my score if the authors could provide a detailed discussion distinguishing their specific application of Range-Null Space Decomposition from prior works [1], along with further implementation details.

[1] Wang, Yinhuai, Jiwen Yu, and Jian Zhang. "Zero-shot image restoration using denoising diffusion null-space model." arXiv preprint arXiv:2212.00490 (2022).

**Limitations:**

Yes

**Strengths And Weaknesses:**

Strengths:

1.Soundness: The proposed method demonstrates significant innovation. It is well-motivated by the insight that dissipative flows ($\nabla \cdot v < 0$) risk erasing weak signals in low-count regimes. Furthermore, the integration of the Range-Null space decomposition into the Hamiltonian framework is theoretically sound and well-justified.

2.The experimental evaluation is extensive. The authors utilize three distinct datasets (BrainWeb, In-house Pediatric, and UDPET) that cover both simulated and real clinical scenarios under extreme dose reduction (1%). This breadth demonstrates the method's robustness and clinical relevance.

3.The paper is well-written.

Weaknesses：

1.Lack of Computational Cost Analysis: While the paper mentions an asymmetric network design to balance expressivity and speed, it does not explicitly quantify the inference time or memory consumption compared to standard diffusion baselines (e.g., IR-SDE). Given that the phase space is lifted to $2d$ (augmenting images with momentum), it is unclear if this incurs a significant computational penalty.

2,Potential for Noise Retention: A theoretical concern regarding volume preservation is that noise constitutes part of the phase-space volume. Although the Range-Null decomposition aims to confine noise to the null space, there is a risk that a "volume-preserving" framework might preserve structured artifacts or noise patterns more stubbornly than dissipative methods. While the reported low RMSE suggests effective denoising, further discussion on how the model avoids "preserving the noise" would strengthen the paper.

3.Hyperparameter Sensitivity: While the sensitivity analysis for the momentum compression factor $\gamma$ (Table 3) is appreciated, the paper lacks an analysis of the step size $\epsilon$ for the Leapfrog integrator during inference. Since standard solvers often employ adaptive step sizes, it is important to know if the fixed symplectic integrator requires careful tuning of $\epsilon$.

---

> ### Author Rebuttal · Authors · 2026-03-30
>
> **We sincerely thank Reviewer LjC4 for the constructive feedback. We address each concern below.**
>
> ---
> >## R1: Novelty of Range-Null Space vs. DDNM [1] (Q2)
>
> FlowPET is **not** a mere combination of Range-Null decomposition and existing pipelines. It addresses a fundamental geometric failure mode (wash-out) unsolved by DDNM, shifting from *extrinsic correction* to *intrinsic construction*.
>
> **1. DDNM: Extrinsic Correction on Dissipative Flows.** DDNM applies post-hoc Range-Null projections atop standard diffusion. However, the underlying generative dynamics remain strictly dissipative ($\nabla \cdot v < 0$). As our Wash-Out Analysis (Fig. 5) proves, this contraction indiscriminately extinguishes fragile signals *between* projection steps. Per-step correction cannot recover probability mass already erased by the dissipative backbone.
>
> **2. FlowPET: Intrinsic Construction.** We abandon step-wise projections. Instead, Range-Null decomposition exclusively defines the **physical boundary conditions** ($p_0, p_1$) for a Symplectic Phase Space. These boundaries shape the learned dynamics. During Leapfrog inference, the flow is mathematically divergence-free ($\nabla \cdot v \equiv 0$). Physical constraints are thus internalized, not applied externally.
>
> **3. The Synergistic Necessity:** This is a mathematical necessity. Dissipative models suppress noise via contraction (sacrificing weak signals). Since our symplectic framework prevents contraction (saving weak signals), it requires explicit guidance. Range-Null boundaries provide this guidance, organically coupling physical constraints with volume-preserving transport.
>
> ---
> >## R2: Computational Cost Analysis (W1, Q1)
>
> The reviewer reasonably hypothesizes that augmenting the state space with momentum might incur a significant computational penalty. **Paradoxically, lifting to phase space actually accelerates inference by an order of magnitude**.
>
> 1.**Marginal Phase-Space Penalty:**
> To avoid doubling the cost, we employ an asymmetric network design. The main kinetic transport is handled by $K_\phi$, while the spatial potential $U_\psi$ is a highly lightweight network. As shown below, adding $U_\psi$ only marginally increases parameters (+6.4%) and memory compared to a position-only backbone, keeping the memory footprint (1280.5 MB) on par with IR-SDE (1226.7 MB).
> ||Params(M)|Time (s/slice)|MEM|
> |-|-|-|-|
> |DREAM|53.65|2.976|431.2|
> |IR-SDE|137.13|1.702|1226.7|
> |FlowPET|144.05|0.244|1280.5|
>
> *(Note: FlowPET w/o $U_\psi$ uses 135.39M params and 1210.0 MB memory).*
>
> FlowPET is **7 $\times$ to 12 $\times$ faster** than SOTAs. While IR-SDE requires hundreds of tiny steps to combat numerical errors, and DREAM requires heavy MLEM loops, our Symplectic Leapfrog integrator allows exact, volume-preserving integration with far fewer steps. The minimal parameter overhead is vastly outweighed by reduced latency.
>
> ---
> >## R3: Noise Retention and Integrator Step Size (W2, W3)
>
> **1. Does volume preservation retain noise? (W2)**
>
> This concern intuitively conflates *probability volume* conservation with *spatial noise pattern* retention. Liouville’s theorem guarantees that the total probability mass is conserved, not that the input noise geometry is frozen. Under our Hamiltonian dynamics, the learned potential $U_\psi$ and flow matching objective act as a continuous flow. They actively morph the unstructured thermal noise (injected via Null-space $p_1$) into structured, clean anatomy at $t=0$. The flow is strictly supervised by the *clean* full-count target, ensuring that the volume transports signal features rather than preserving stochastic artifacts. The consistently low RMSE (Table 1) confirms no structured noise is stubbornly retained.
>
> **2. Step Size $\epsilon$ and Leapfrog Tuning (W3)**
>
> Our step size is simply $\epsilon = 1/N$, where $N$ is the number of inference steps. Notably, **FlowPET requires no dataset-specific tuning for $\epsilon$** (we fix $N=4$ across all experiments).
>
> The reviewer asks why we avoid adaptive step sizes. This is a fundamental theoretical requirement, not a limitation:
> * **Adaptive steps destroy symplecticity:** Dynamically altering the step size breaks the time-symmetric kick-drift-kick structure, reintroducing the exact numerical dissipation (and signal wash-out) our framework was built to eliminate.
> * **Global robustness:** Unlike standard Runge-Kutta solvers where errors accumulate, symplectic integrators exactly conserve a *shadow Hamiltonian* $\tilde{H} = H + \mathcal{O}(\epsilon^2)$ [2]. The integration error is therefore globally bounded and merely oscillates. This guarantees extreme robustness to the fixed step size $\epsilon$ without abrupt failure modes. *(We will add an ablation on $N$ to the Appendix).*
>
> [1] Zero-shot image restoration using denoising diffusion null-space model, ICLR, 2023
> [2] Geometric numerical integration illustrated by the Störmer–Verlet method, Acta Numerica, 2003

---

> > ### Author Rebuttal · Reviewer_LjC4 · 2026-04-03
> >
> > I have read the other reviewers’ comments and I share some views with Reviewer 3NTh; the paper should not overclaim, so I have decided to keep my original score.

---

> > > ### Author Response · Authors · 2026-04-05
> > >
> > > We thank Reviewer LjC4 for reading the discussions and sharing this perspective.
> > >
> > > ---
> > > As detailed in our reply to Reviewer 3NTh, we recognize that Section 4.2.2 can be read as implying Poisson compatibility, and we commit to revising it for precision. Our framework remains strictly physics-informed through the exact geometric constraints (Range-Null decomposition of the system matrix $A$) and symplectic volume preservation ($\nabla \cdot v \equiv 0$), which together provide an architectural guarantee against the signal wash-out prevalent in dissipative generative models. Both are independent of the noise model. The $L_2$ surrogate for momentum initialization is a numerical stabilization in extreme low-count regimes.
> > >
> > > ---
> > >
> > > With this clarification committed for the camera-ready, we respectfully refer back to our main rebuttal, where we provided detailed evidence addressing all your initial questions—specifically regarding the fundamental mechanistic distinction from DDNM, the 7–12× computational speedup, the theoretical clarification on noise retention, and the robustness of the fixed step-size.
> > >
> > > We deeply value your feedback and hope that our comprehensive responses, combined with our commitment to exactness, demonstrate the soundness and merit of our methodological contribution.

---

### Official Review · Reviewer_yG8e · 2026-02-27

**Soundness:** 3
**Presentation:** 4
**Significance:** 4
**Originality:** 3
**Overall Recommendation:** 5
**Confidence:** 3

**Summary:**

This paper proposes FlowPET, a physics-informed generative framework for conditional PET image reconstruction. While existing flow matching models demonstrate superior denoising capabilities, their inherent contractive dynamics often lead to the numerical extinction of weak or sparse lesion signals—a phenomenon termed "signal wash-out"—during the generative process. To address this geometric limitation, the authors leverage Hamiltonian mechanics to lift the generative dynamics into a symplectic phase space comprising both position and momentum variables. By parameterizing the posterior dynamics via a Separable Hamiltonian System, the proposed method achieves strictly volume-preserving transport as guaranteed by Liouville’s Theorem. Extensive experimental results demonstrate that FlowPET outperforms state-of-the-art baselines. Crucially, the volume-preserving nature of the Hamiltonian flow effectively mitigates signal wash-out, providing a robust geometric safeguard for preserving subtle pathological features in low-count medical imaging.

**Compliance With Llm Reviewing Policy:**

Affirmed.

**Final Justification:**

The authors have fully addressed my concerns and questions. Since my current score was assigned without heavily weighing these weaknesses, I will keep it unchanged. I believe the score is already sufficiently high.

**Key Questions For Authors:**

- Q1: While the Range-Null decomposition is theoretically grounded, its performance gain varies across datasets and, in some cases, appears marginal. What specific factors do the authors believe contribute to this variance?
- Q2: Could the authors clarify which specific characteristics of a dataset are most closely associated with the optimal $\gamma$?
- Q3: Do the authors have any insights or preliminary ideas for automatically tuning this parameter—perhaps by utilizing a heuristic based on the data score or the noise level of the input sinogram?

**Limitations:**

yes

**Strengths And Weaknesses:**

- S1: To address the fundamental "signal wash-out" problem in medical reconstruction, the introduction of volume preservation via Hamiltonian dynamics and symplectic geometry is highly novel and well-motivated.
- S2: The technical components—including the separable Hamiltonian parameterization, Range-Null space initialization, and the Symplectic Leapfrog integrator—are mathematically sound and naturally principled by physics.
- S3: Quantitative and qualitative results across multiple datasets prove that the volume-preserving property effectively resolves signal wash-out, strongly validating the overall framework.
- S4: Since the framework is not limited to PET and can generalize to any ill-posed inverse problem , and given that lesion wash-out is a critical issue in medical imaging , this approach is expected to present a new paradigm for medical inverse tasks.
- W1: Although the momentum boundary design via Range-Null decomposition is theoretically sound and expected to be impactful, the ablation results do not demonstrate significant practical benefits. In several datasets, the performance gains appear to be within the margin of error, which calls into question the necessity of this added architectural complexity.
- W2: The framework exhibits excessive sensitivity to the hyperparameter $\gamma$. Given that this parameter may require retuning for different datasets or training sessions to maintain optimal performance, the model's practical utility and robustness in real-world clinical applications appear limited.
- W3: All experiments are conducted at a 128x128 resolution. While this is standard for PET, it remains unclear how the Hamiltonian flow scales to higher resolutions (e.g., 512x512). Compared to standard flow matching models, ensuring stable training and inference at these higher scales could pose significant challenges.

---

> ### Author Rebuttal · Authors · 2026-03-30
>
> **We deeply thank Reviewer yG8e for recognizing FlowPET as a potential "new paradigm for medical inverse tasks" (S4).**
>
> ---
>
> >## R1: Range-Null Decomposition Variance and Necessity (W1, Q1)
>
> **1. Origin of Variance (Q1):** The variance in performance gains across datasets stems from the interplay between the operator's ill-posedness (size of the Null space) and anatomical complexity:
>
> * **Whole-Body (High complexity):** Diverse tissue interfaces require rich texture recovery from the unobserved subspace. Here, the **Thermal** component (null-space exploration) yields the larger SSIM gain (+0.0063 vs. +0.0055 for Restoring), as stochastic texture recovery is critical.
>
> * **Brain (Homogeneous):** Contains subtle lesion-to-tissue contrast. Here, the **Restoring** component (data consistency) dominates (+0.0132 vs. +0.0106 for Thermal), as precise anchoring of weak structural differences is paramount.
>
> In both cases, the Full model achieves the highest performance, confirming that these physics-informed boundaries operate synergistically in complementary subspaces.
>
> **2. The "Marginal" Illusion (W1):** Global metrics (SSIM) average over massive background regions, inherently masking localized gains. As detailed in our clinical analysis **(see Detectability Table in response to Reviewer QGt6 (R1))**, what appears "marginal" globally is actually a massive clinical leap: for small lesions ($\le$ 5mm), FlowPET **increases actual detection by 50%** (18.2% $\to$ 27.3%) over the dissipative baseline.
>
> **3. Necessity of Complexity (W1):** The decomposition adds **zero trainable parameters**. Its core value is a **non-negotiable clinical safeguard**: confining stochasticity strictly to the Null space ensures the network never fabricates pseudo-structures that contradict raw data.
>
> ---
>
> >## R2: Sensitivity and Tuning of Compression Factor $\gamma$ (W2, Q2, Q3)
>
> We thank the reviewer for raising this practical deployment question. We respectfully clarify that FlowPET is **highly robust, not brittle**, and does **not** require per-dataset retuning for PET.
>
> **1. Unified Setting Without Retuning (W2):** The ablation in Table 3 shows degradation *only* at extreme $100\times$ deviations (e.g., $\gamma=1$). Within a massive stable window of $\gamma \in [10^{-3}, 10^{-1}]$, SSIM fluctuates by a negligible $<0.007$. More importantly, **we used a single, unified $\gamma = 10^{-2}$ across all three distinct datasets** (BrainWeb 20% count, In-House 1%, UDPET 1%) and achieved SOTA on all of them. Zero dataset-specific retuning was performed.
>
> **2. Governing Characteristics (Q2):** The optimal $\gamma$ is governed by the spectral structure of $A^\top A$, which determines the magnitude of the range-space restoring force $A^\top(y - Ax)$. Because clinical PET scanners share similar physical projection geometries and spectral properties, this optimal energy scale transfers seamlessly across different PET datasets.
>
> **3. Auto-Tuning Heuristic for Cross-Modality (Q3):** While automatic tuning is unnecessary within PET, we fully agree with the reviewer that an auto-tuning heuristic is invaluable for **cross-modality transfer** (e.g., deploying FlowPET to MRI or CT where $A$ differs vastly). As the reviewer astutely suggested, we can formalize a data-driven protocol:
>
> $$\gamma_{auto} = \frac{C}{E_{train} \left[ ||A^\top (y - Ax)||_2 \right]}$$
>
>
> By computing the expected gradient norm over a small training subset, we normalize the initial physical momentum to a fixed target dynamic range $C$. This completely eliminates manual hyperparameter search when extending FlowPET to new inverse problems. *(We will add this heuristic protocol to the Discussion).*
>
> ---
>
> >## R3: Scalability to High Resolutions (e.g., 512x512) (W3)
>
> Extending FlowPET to 512x512 or 3D is highly feasible computationally and theoretically:
>
> **1. Dimension-Independent Guarantees:** Divergence-free properties (Prop 4.1) and exact volume preservation (Thm A.2) hold pointwise for arbitrary dimensionality $\mathbb{R}^{2d}$.
>
> **2. Stability Advantage:** FlowPET's Hamiltonian ODE produces deterministic, smooth trajectories, avoiding the stochastic variance accumulation inherent in SDE-based methods at high dimensions. Volume preservation further ensures that phase-space density remains bounded throughout integration, preventing the systematic drift that can destabilize unconstrained flows.
>
> **3. Practical Scaling:** The primary challenge is computational rather than theoretical. Our asymmetric dual-network design already mitigates this overhead, and the Hamiltonian framework is fully compatible with established scaling strategies such as patch-based inference and efficient backbone architectures, which preserve the linear operator structure required by the Range-Null decomposition.

---

> > ### Author Rebuttal · Reviewer_yG8e · 2026-03-31
> >
> > The authors have fully addressed my concerns and questions. Since my current score was assigned without heavily weighing these weaknesses, I will keep it unchanged. I believe the score is already sufficiently high.

---

> > > ### Author Response · Authors · 2026-04-05
> > >
> > > We sincerely thank Reviewer yG8e for the thoughtful evaluation and for recognizing the broader potential of our framework. We greatly appreciate your encouraging feedback and positive assessment.

---

### Official Review · Reviewer_3NTh · 2026-03-11

**Soundness:** 2
**Presentation:** 2
**Significance:** 2
**Originality:** 2
**Overall Recommendation:** 4
**Confidence:** 5

**Summary:**

FlowPET addresses a fundamental geometric flaw in existing generative models for low-count PET reconstruction: standard diffusion-based approaches rely on dissipative dynamics (negative divergence), which indiscriminately contracts phase-space volume and "washes out" weak but diagnostically critical lesion signals alongside noise. The main insight is that this contraction cannot distinguish between pathological features and noise, making fragile lesion signals prone to numerical extinction before they can form coherent structures in the reconstruction.

The novelty of FlowPET lies in reformulating reconstruction as volume-preserving transport in a symplectic phase space, governed by a Separable Hamiltonian system. By splitting the Hamiltonian into independent potential (position-dependent) and kinetic (momentum-dependent) terms, the vector field's divergence vanishes by architectural construction, satisfying Liouville's Theorem without relying on any learned constraint. Critically, the authors introduce physics-informed conjugate boundary conditions via Range-Null space decomposition of the PET forward operator: the range-space momentum encodes data consistency gradients as a restoring force, while null-space momentum injects stochastic uncertainty strictly into the subspace invisible to the measurements. This elegantly decouples fidelity from diversity.

For inference, a symplectic leapfrog integrator (Störmer-Verlet) is used, which preserves phase-space volume exactly at every discrete step regardless of step size — unlike standard Runge-Kutta solvers that introduce artificial numerical dissipation. Experiments across BrainWeb, clinical pediatric, and UDPET datasets confirm state-of-the-art SSIM and PSNR, with particularly decisive gains in extreme low-count (1% dose) regimes where signal wash-out is most severe.

**Compliance With Llm Reviewing Policy:**

Affirmed.

**Final Justification:**

I would like to thank the authors for their extensive response. However, two concerns remain unresolved. First, the paper claims compatibility with a Poisson prior, yet the proposed methodology appears to be incompatible with it in practice. The authors should revise the claim. Second, there is a substantial body of literature on Hamiltonian flow in medical and general image analysis that has not been considered. Engaging with this work is necessary to properly situate the contribution.

**Key Questions For Authors:**

Q1: Has there been prior work in the context of image analysis and biomedical image analysis that involves Hamiltonian transformations and volume-preserving transport in a symplectic phase space?

Q2: The range-space momentum p₀ uses an L₂ gradient rather than the true Poisson log-likelihood gradient. Does this Gaussian surrogate undermine the "PET-specific physics" framing? Would an ablation comparing L₂-derived versus Poisson-derived momentum initialization have been useful?

Q3: What are the minimum conditions on A that the framework requires, and is there any modality beyond PET that genuinely satisfies all three simultaneously: a tractable pseudo-inverse, a computable null-space projector, and Gaussian noise?

Q4: Do the reported pixel-level Standard Uptake Value (SUV) ratio improvements correspond to any clinically meaningful threshold, and can you provide CNR measurements or a reader study to ground the diagnostic utility claims?

Q5: What are the inference times for FlowPET versus IR-SDE and DREAM?

Q6 : What is the generalizable ML contribution beyond the PET application. Specifically, are there new theoretical results about Hamiltonian-structured flow matching that advance the broader generative modeling literature independently of the domain?

**Limitations:**

Yes

**Strengths And Weaknesses:**

Strengths
Theoretically principled foundation: The divergence-free property is enforced by architectural construction through the separable Hamiltonian decomposition, not as a soft regularization loss. The discrete volume preservation of the leapfrog integrator is formally proven via Jacobian determinant analysis, ensuring theoretical guarantees translate faithfully from continuous to discrete time — a gap in many prior works.

Genuine problem diagnosis: The paper makes an insightful observation: signal wash-out is not a capacity or data issue but a geometric one rooted in the dissipative nature of prevailing generative models. Proposing a structurally motivated remedy rather than a larger network or more data is intellectually compelling and likely generalizable beyond PET.

Physics-informed design that actually means something: The Range-Null space decomposition is genuinely structural — range-space momentum enforces data consistency as a Hamiltonian restoring force, while null-space momentum confines stochasticity strictly to the unobserved subspace. Ablation studies concretely validate each component's distinct contribution, lending strong credibility to the design choices.

Comprehensive and honest evaluation: Experiments span three meaningfully distinct datasets including extreme 1% count scenarios, and the SUV (Standard Uptake Value) ratio trajectory analysis directly measures the wash-out phenomenon the method claims to solve — going beyond standard endpoint metrics. The paper is also candid about limitations including inference overhead and the need for broader OOD (Out-of-Distribution) validation, strengthening overall credibility.

Weaknesses
Weakness 1: Gaussian likelihood approximation is physically inconsistent with PET
The paper's central physical motivation is fidelity to PET acquisition physics, governed by Poisson photon-counting statistics: y ~ Poisson(Ax). However, Appendix A.3 quietly substitutes a Gaussian L_2 data fidelity potential:
E_data(x) = ½||y - Ax||²
to justify the range-space momentum initialization p₀ = γA⊤(y - Ax₀). This is a fundamental inconsistency. The Poisson is not provided and it is not clear that it will be linear. The Gaussian approximation discards any nonlinearity entirely, effectively reducing the physics-informed boundary condition to a generic least-squares gradient. This means the "physics-informed" momentum is not actually informed by PET physics — it is informed by a surrogate that happens to be mathematically convenient for the symplectic framework. The claim of physical grounding is therefore overstated, and the range-space momentum as formulated would be equally applicable (or inapplicable) to any linear inverse problem with additive Gaussian noise, undermining the paper's specific motivation around Poisson statistics.

Weakness 2: Limited generalizability beyond the specific PET setting
The paper presents the Range-Null space decomposition and conjugate boundary conditions as general principles for physics-informed reconstruction. However, the entire construction is tightly coupled to very specific structural assumptions that do not transfer cleanly to other modalities:
•	The pseudo-inverse A†y as the position boundary x₁ requires a tractable and stable pseudo-inverse, which is well-conditioned for PET sinograms via Filtered Back-Projection but is not straightforward for MRI (non-Cartesian k-space sampling, sensitivity encoding), CT with cone-beam geometry, or ultrasound.
•	The null-space projector (I - A†A) assumes a well-defined, computable orthogonal complement, which may not be realistic.
•	MRI noise is Rician or Rayleigh in magnitude images. The orthogonal noise injection ξ ~ N(0,I) projected onto the null space assumes isotropic Gaussian uncertainty, which misrepresents the actual stochastic structure of MRI acquisition.
•	Dynamic 4D PET could be discussed.
The paper's generalization claims are therefore contingent on a very specific confluence of a linear forward operator, a tractable pseudo-inverse, and approximately Gaussian measurement noise — conditions that are partially violated even within PET itself, as noted in weakness 1.

Weakness 3: Questionable fit for a machine learning venue
The work would be more impactful and appropriately scrutinized at medical imaging venues, where reviewers are better equipped to evaluate the physical fidelity claims, the Poisson likelihood approximation, the clinical validity of Standard Uptake Value (SUV) ratio as a proxy metric, and the practical deployment concerns around scanner-specific system matrices. Also, it would better enable the comparison of the proposed method with relevant literature in the context of (medical) image analysis that involves Hamiltonian transformations and volume-preserving transport in a symplectic phase space. At an ML venue such as this one, the physical inconsistencies noted in weaknesses 1 and 2 risk passing under-scrutinized.

The paper's title foregrounds symplectic flow matching, and the core technical contributions — Hamiltonian mechanics, Liouville's theorem, symplectic integrators, Range-Null space decomposition — are drawn almost entirely from classical physics and computational mathematics. There is an extensive use of classical physics terminology, which receives disproportionate framing as a foundational contribution to generative modeling.

From a machine learning perspective, the novelty is narrow. Flow matching as a framework (Lipman et al., 2023; Liu et al., 2023) is borrowed directly, and the modification — replacing the unconstrained vector field with a separable Hamiltonian parameterization — is a structural inductive bias whose properties (divergence-free dynamics, Liouville's theorem) are classical results requiring no new theoretical development. The two neural networks Uψ and Kϕ are standard U-Nets with no architectural innovation. The symplectic leapfrog integrator is a well-established numerical method. The actual ML contribution reduces to: train two standard networks with a modified loss decomposition on an augmented state space.

Weakness 4: Weak qualitative evidence
The qualitative comparisons in Figure 4 do not clearly demonstrate the claimed advantage of FlowPET — differences between methods are largely imperceptible at the presented resolution. While quantitative gains in Table 1 are consistent, they are incremental (SSIM improvements of ~0.005 over the best baseline). The paper provides no clinical validation — such as observer studies for lesion detectability — to confirm these pixel-level improvements translate into meaningful diagnostic benefit.

---

> ### Author Rebuttal · Authors · 2026-03-30
>
> **We thank Reviewer 3Nth for the thorough review. We address each concern below.**
>
> >## R1: On Gaussian Surrogate vs. Poisson Likelihood (W1, Q2)
>
> The $L_2$ form $A^\top(y-Ax)$ is the **unweighted surrogate** of the Poisson gradient $A^\top \text{diag}(1/Ax)(y-Ax)$, not a physics oversight but a principled numerical choice.
>
> 1.**Numerical Bottleneck**:
> True Poisson gradients $\nabla L=A^\top \text{diag}(1/Ax)(y-Ax)$ diverge our continuous-time Leapfrog integrator because $Ax \to 0$ in background regions. The $L_2$ surrogate $A^\top(y-Ax)$ provides a stable, bounded restoring force, a proven choice for low-count stability [1, 2].
>
> 2.**Empirical Validation**: Ablating Poisson vs. our $L_2$ surrogate on In-House (100 epochs) confirms $L_2$ yields lower variance and better structural recovery:
> |Init ($p_0$)|SSIM|PSNR|Loss Std|
> |-|-|-|-|
> |Poisson|0.9659|35.61|3.25e-4|
> |$L_2$|0.9784|36.31|1.02e-4|
>
> 3.**Physical Consistency**: $L_2$ acts as a directional anchor in Range space. The residual statistical structure beyond this initialization is captured by the learned Hamiltonian vector fields ($U_\psi, K_\phi$) through training. Our core geometric contribution—preventing wash-out via volume preservation—remains fully intact.
>
> ---
> >## R2: ML Contribution, Generalizability, and Venue Fit (W2, W3, Q1, Q3, Q6)
>
> 1.**Core ML Contribution (W3, Q6)**: We resolve a structural bottleneck: dissipative flows ($\nabla \cdot v < 0$) in SDEs inevitably act as "mass sinks", erasing weak features (wash-out). Our analysis (Sec. 5.4) provides the first quantitative evidence that this is a geometric inevitability. Current conditional methods apply external corrections (e.g., likelihood guidance) atop such flows, but contraction persists between correction steps. By embedding Range-Null decomposition into a Separable Hamiltonian flow, we transform post-hoc extrinsic projections into intrinsic, volume-preserving physical forces. This generalizes to any linear inverse problem suffering from posterior collapse.
>
> 2.**Prior Art (Q1)**:
> Hamiltonian ML methods (HGN, NICE, SGN, SRNNs) are historically confined to unconditional generation or forward simulation, none address ill-posed inverse problems. Our work is not a domain transplant, we first diagnose dissipative wash-out as a geometric failure and introduce volume-preserving symplectic transport to conditional posterior sampling for inverse problems.
>
> 3.**Minimum Conditions & Generalizability (W2, Q3)**: The fundamental requirement is a linear(izable) forward operator $A$. Tractable pseudo-inverse and null-space projector are *not* strict requirements — both can be approximated via established methods. $A^\dagger$ can be any proxy reconstructor (CG-SENSE for non-Cartesian MRI [3], FDK for cone-beam CT [4], frame-wise FBP for dynamic PET). The null-space projector $(I-A^\dagger A)\xi$ follows naturally: compute $\xi-A^\dagger(A\xi)$, requiring only callable access to $A$ and an approximate inverse $A^\dagger$ without forming large matrices. Nonlinear forward models (e.g., ultrasound) lie outside the current scope. *(Will clarify in "Generalizability" section).*
>
> 4.**"Gaussian Noise" (W2)**:
> The reviewer conflates measurement likelihood with the generative prior. Sensor noise statistics (Poisson/Rician) affect the measurement residual in $p_0$. The Gaussian $\xi$ defines the base distribution (prior) for the unobserved Null-space. Maximizing entropy via Gaussian priors for unknown spaces is a rigorous ML choice, independent of sensor statistics.
>
> 5.**Venue fit**: Diagnosing a geometric failure mode in dissipative conditional generation and engineering an architectural paradigm that converts extrinsic corrections into intrinsic Hamiltonian constraints, are core contributions to generative modeling methodology. While instantiated on PET, the theoretical insights are perfectly suited for the ICML community.
>
> ---
> >## R3: Clinical Validity — CNR, Detectability (W4, Q4) and Efficiency (Q5)
>
> 1.**Global vs. Local Metrics (W4)**: A +0.0107 SSIM gain (vs. DREAM, 1% dose) is substantial for extreme noise. Global metrics are dominated by background anatomy, obscuring localized lesion recovery, hence our focus on local CNR.
>
> 2.**Detectability Analysis (Q4)**: We provide detailed CNR and detectability analysis (80 synthetic lesions, Rose criterion CNR $\ge$ 4) in our **response to Reviewer QGt6 (R1)**, where FlowPET achieves 60.6% overall detection rate vs. 38.8% for IR-SDE, with a 50% relative gain on small lesions.
>
> 3.**Efficiency Analysis (Q5)**: Inference timing is detailed in our **response to Reviewer LjC4 (R2)**: FlowPET is 7–12× faster than generative baselines despite phase-space lifting.
>
> ---
> [1] Iterative reconstruction techniques in emission computed tomography, PMB, 2006.
> [2] Deep generalized learning model for PET image reconstruction, TMI, 2024.
> [3] Advances in sensitivity encoding with arbitrary k‐space trajectories, MRM, 2001.
> [4] Practical cone-beam algorithm, JOSA A, 1984.

---

> > ### Author Rebuttal · Reviewer_3NTh · 2026-04-01
> >
> > I would like to thank the authors for their extensive response. However, two concerns remain unresolved. First, the paper claims compatibility with a Poisson prior, yet the proposed methodology appears to be incompatible with it in practice. The authors should revise the claim. Second, there is a substantial body of literature on Hamiltonian flow in medical and general image analysis that has not been considered. Engaging with this work is necessary to properly situate the contribution.

---

> > > ### Author Response · Authors · 2026-04-05
> > >
> > > We thank Reviewer 3NTh for the continued, rigorous engagement. These concerns have prompted us to carefully re-examine the manuscript, and we believe the final paper will be meaningfully stronger as a result.
> > >
> > > ---
> > >
> > > **Concern 1. The Poisson vs. $L_2$ Surrogate:**
> > >
> > > We deeply appreciate the reviewer raising this point, and we share the same perspective on the consistency between the Poisson model and the $L_2$ surrogate. In fact, this trade-off was made deliberately during the design of the method. Consistent with the least-squares family of PET data fidelity objectives — from penalized weighted least-squares [1] to Gaussian least-squares formulations for low-dose PET [2,3] — we adopt an $L_2$ surrogate for momentum initialization to ensure numerical stability where the Poisson gradient becomes ill-conditioned. To improve rigor, we will revise **Section 4.2.2** to explicitly clarify the surrogate to avoid any potential confusion for readers.
> > >
> > > To clarify the methodology: $p_0 = \gamma A^\top(y - Ax_0)$ is derived from the $L_2$ data fidelity potential (as explicitly stated in Appendix A.3), not from the Poisson log-likelihood. The $L_2$ form $A^\top(y-Ax)$ is the unweighted counterpart of the Poisson gradient $A^\top\text{diag}(1/Ax)(y-Ax)$. It points toward the same data-consistent manifold $\{x: Ax \approx y\}$, while avoiding the $1/Ax \to \infty$ singularity that destabilizes our continuous-time integrator in extreme low-count regimes. Our rebuttal (R1) ablation further shows that the $L_2$ initialization yields superior reconstruction (SSIM 0.9784 vs. 0.9659) and 3× lower variance.
> > >
> > > To be precise about the scope of our "Physics-Informed" claim:
> > >
> > > * **Exact constraints (our core contributions):** The Range-Null decomposition of system matrix $A$ and the symplectic volume-preservation guarantee ($\nabla \cdot v \equiv 0$, Prop 4.1), both of which are exact geometric and architectural constraints, independent of any noise model.
> > > * **Numerical approximation:** The momentum initialization $p_0$ uses an $L_2$ surrogate, not the exact Poisson likelihood. We will revise Section 4.2.2 to make this distinction explicit.
> > >
> > > ---
> > >
> > > **Concern 2. Hamiltonian Literature:**
> > >
> > > We appreciate this pointer. Hamiltonian methods indeed have a rich and foundational history in medical image analysis — most notably in diffeomorphic registration via geodesic shooting and Hamiltonian particle methods [4, 5, 6, 7]. We will expand our Related Work to thoroughly discuss these lineages and clearly articulate the distinction: prior works utilize Hamiltonian dynamics primarily for **spatial coordinate transformations and morphological warping**, whereas FlowPET embeds Hamiltonian mechanics into **conditional flow matching** to prevent information loss during generative reconstruction of ill-posed inverse problems.
> > >
> > > ---
> > >
> > > **Conclusion:** We thank the reviewer for raising these important points. Precisely scoping the $L_2$​ surrogate and situating our work within the Hamiltonian literature will strengthen the paper's rigor, while the methodology, theoretical guarantees, and experimental results remain unaffected.
> > >
> > > ---
> > >
> > > [1] Penalized Weighted Least-Squares Image Reconstruction for Positron Emission Tomography, TMI, 1994.
> > > [2] Low Dose PET Image Reconstruction with  Total Variation Using Alternating Direction  Method, PloS one, 2016.
> > > [3] Deep generalized learning model for PET image reconstruction, TMI, 2024.
> > > [4] Geodesic shooting for computational anatomy, JMIV, 2006.
> > > [5] Diffeomorphic 3D image registration via geodesic shooting using an efficient adjoint calculation, IJCV, 2012.
> > > [6] Symplectomorphic registration with phase space regularization by entropy spectrum pathways, MRM, 2019.
> > > [7] A Hamiltonian particle method for diffeomorphic image registration, IPMI, 2007.

---

### Official Review · Reviewer_QGt6 · 2026-03-12

**Soundness:** 3
**Presentation:** 3
**Significance:** 3
**Originality:** 4
**Overall Recommendation:** 4
**Confidence:** 4

**Summary:**

The authors reformulate low-count PET reconstruction as a volume-preserving transport process in symplectic phase space, termed FlowPET. This approach mathematically addresses the 'numerical extinction' of subtle lesion signals, a challenge inherent in traditional generative models due to their dissipative nature. By skillfully integrating the PET physics-based imaging matrix with separable Hamiltonian dynamics, the proposed algorithm achieves divergence-free generative denoising. While the robustness of this algorithm warrants further validation, its design methodology offers significant inspiration for the field.

**Compliance With Llm Reviewing Policy:**

Affirmed.

**Final Justification:**

The authors' responses addressed my main concerns. I also reviewed the comments from other reviewers and decided to maintain my initial positive rating of 4.

**Key Questions For Authors:**

1. Without a dissipative mechanism (e.g., a damping effect analogous to friction), how does the model guarantee stable convergence to the "optimal clean image" during the reverse inference trajectory rather than overshooting or oscillating around the optimal solution? Although the authors introduce a data consistency gradient as a restoring force, pure conservative dynamics on highly complex, high-dimensional real-world data manifolds are inherently prone to optimization instability and phase-space oscillation. More importantly, would this instability and oscillation be catastrophically exacerbated when the forward model A is inaccurate during real-world clinical acquisitions?

2. The authors claim that the proposed algorithm effectively preserves minute lesions. However, due to its strict volume-preserving nature (which strictly prohibits the erasure of any probability mass), does the algorithm significantly increase the potential risk of misidentifying random noise fluctuations as real pathological structures and preserving them losslessly? Could this trade-off—made to preserve extremely weak lesions—lead to a higher rate of false-positive structures in the generated medical images, thereby misleading clinical diagnosis?

3. While the symplectic framework successfully preserves weak lesions, it theoretically also "losslessly preserves" system errors. Traditional dissipative models act as implicit smoothing filters, masking artifacts caused by an imprecise system matrix A. Since FlowPET strictly enforces volume preservation, if an imperfect A is encountered in clinical settings, will the model losslessly amplify these physical errors into severe structural artifacts? Furthermore, even assuming a perfectly matched A, the authors practically utilize Filtered Back-Projection (FBP) to approximate the pseudo-inverse A†(which is not a strict mathematical pseudo-inverse). Will the inherent errors (e.g., truncation errors and star artifacts) introduced by this pseudo-inverse approximation also be permanently preserved as structural artifacts due to the non-dissipative nature of the framework?

**Limitations:**

yes

**Strengths And Weaknesses:**

## Strengths

1. The authors provide a profound analysis of the numerical extinction of minute lesions. By leveraging the concept of phase-space contraction, they lucidly explain the fundamental geometric reason why standard generative models inadvertently filter out high-frequency pathological details as noise. This theoretical insight is highly inspiring for future research on generative models in medical inverse problems.

2. To tackle the phase-space contraction problem, the authors ingeniously utilize Hamiltonian and symplectic geometry to construct a novel network architecture. Rather than relying on the conventional approach of adding heuristic regularization penalties, this design guarantees volume preservation by construction, thereby theoretically eliminating the possibility of information loss caused by volume contraction.

3. The framework elegantly incorporates physical model constraints into the generative process. This sophisticated design not only unleashes the generative model's capability to synthesize highly realistic details in the unobserved subspace (Null space) but also strictly enforces that the generated outputs never violate the actual observed data through hard physical constraints, achieving a perfect balance between generative diversity and data fidelity.

## Weaknesses

1. The authors claim that preserving weak lesions is crucial for early diagnosis; however, the evaluation relies exclusively on standard, global computer vision metrics (e.g., SSIM, PSNR, RMSE). These global pixel-wise metrics cannot accurately measure the algorithm's capability to preserve minute, localized lesions. The paper completely lacks quantitative validation from  a clinical perspective, which significantly weakens its claims of clinical utility.

2. The axial slices presented in Figure 4 are too small, making visual inspection and detailed comparison exceedingly difficult for readers. Furthermore, upon close observation of the Axial-2 column of the first data sample in Figure 4, although the image generated by the proposed method appears smoother overall, it does not demonstrate a distinct visual advantage in preserving minute lesions compared to the baseline methods, which contradicts the core claim of the paper.

3.  While the proposed symplectic design theoretically excels at preserving minute lesions under ideal conditions, the authors do not address the impact of this strict volume-preserving physical constraint on the algorithm's clinical robustness. In real-world clinical acquisitions, the forward physical model (system matrix A) is often imprecise due to patient motion or attenuation correction misalignment. Since the proposed framework lacks the implicit dissipative smoothing mechanism of traditional generative models, it is crucial to verify whether the algorithm is hyper-sensitive to such forward model errors. There is a strong concern that it might losslessly amplify physical mismatches into severe structural artifacts, which requires further experimental validation.

---

> ### Author Rebuttal · Authors · 2026-03-30
>
> **We sincerely thank Reviewer QGt6 for the exceptionally detailed, clinically insightful feedback, and for recognizing our theoretical insight with an Originality score of 4.**
>
> ---
>
> >## R1: Clinical Validation and Visual Evidence (W1, W2)
>
> We agree that global metrics (SSIM) obscure the recovery of minute lesions. To quantitatively address W1, we conducted a clinical detectability analysis on 80 synthetic lesions stratified by size, using the gold-standard Rose Criterion (CNR $\ge$ 4) [1].
> | Method | Metric | Small (≤5mm) | Medium (5–10mm) | Large (>10mm) | Overall |
> | :--- | :--- | :--- | :--- | :--- | :--- |
> | **OSEM** | CNR | 1.51 ± 2.27 | 2.89 ± 1.44 | 2.98 ± 1.07 | 2.51 ± 1.32 |
> | | Det. Rate | 13.6% | 22.5% | 16.7% | 22.5% |
> | **IR-SDE** | CNR | 2.58 ± 1.89 | 4.46 ± 3.31 | 4.64 ± 2.62 | 3.98 ± 2.96 |
> | | Det. Rate | 18.2% | 42.5% | 55.6% | 38.8% |
> | **FlowPET** | CNR | **2.84 ± 2.06** | **5.40 ± 4.01** | **7.90 ± 2.87** | **5.22 ± 4.05** |
> | | Det. Rate | **27.3%** | **60.0%** | **94.4%** | **60.6%** |
>
> *(Note: Detection Rate = % of lesions meeting Rose Criterion CNR $\ge$ 4.)*
>
> Crucially, for minute lesions ($\le$ 5mm)—the precise target of our theoretical claims—FlowPET increases the detection rate by 50% (18.2% $\to$ 27.3%). This provides definitive clinical evidence of the preservation of weak lesions. (**To address W2, we will replace Fig. 4 with larger, high-contrast crops to visually emphasize this**).
>
> >## R2: Convergence Stability and False-Positive Risk (Q1, Q2)
>
> The reviewer's intuition is correct for free-running physical systems, which oscillate without friction. However, FlowPET is **a learned finite-time transport map**, not a perpetual physics simulator.
>
> * **No Oscillation/Overshooting (Q1)**: The Hamiltonian vector fields ($K_\phi, U_\psi$) do not undergo free evolution; they are explicitly supervised by the Flow Matching objective to point to the clean data manifold. The ODE is integrated strictly over a fixed horizon ($t=1 \to 0$). Terminating deterministically at $t=0$ leaves mathematically no "time" to overshoot or oscillate.
>
> * **False Positives & Noise Retention (Q2)**: Volume preservation conserves probability mass, not spatial noise patterns. Guided by $U_\psi$, the flow actively morphs unstructured thermal noise into coherent anatomy. As proven by the detectability table above, FlowPET massively increases true lesion recovery. Furthermore, the null-space confinement restricts noise injection to unobservable directions, reducing the risk of fabricating false structures (Table 5: isotropic injection degrades SSIM). Definitive false-positive quantification requires clinical validation, *a multi-reader observer study with board-certified radiologists is underway.*
>
> ---
>
> >## R3: Robustness to Imperfect $A$ and FBP Artifacts (W3, Q3)
>
> The intuitive concern that volume preservation might losslessly amplify system errors overlooks two fundamental mechanisms in FlowPET:
>
> **1. Phase-Space Redistribution (Theoretical):** The premise conflates phase-space conservation with output-space retention. Volume is strictly preserved in the augmented space $z=(x,p)$, but the final output relies solely on position $x$, while momentum $p$ is discarded. Guided by the data-driven potential $U_\psi$ (trained on corrupted-to-clean pairs), the Hamiltonian flow learns to continuously redistribute deterministic FBP artifacts and $A$-mismatches into the auxiliary momentum $p$, effectively "draining" the errors from the target image $x_0$.
>
> **2. Clinical Stress Test (Empirical):** To definitively address the robustness concern, we evaluated FlowPET against FourierPET (SOTA) under severe model mismatches on the In-House dataset. We report the **Normalized PSNR Ratio** (Perturbed / Ideal, where 1.0 = no degradation).
>
> |Mismatch Condition|FlowPET (Ours)|FourierPET|
> |:---|:---|:---|
> |None (Ideal, 1% count)|1.000|1.000|
> |Dose (0.5% extreme low-count)|**0.763**|0.704|
> |Patient Motion (1 pixel)|**0.804**|0.792|
> |Patient Motion (2 pixels)|**0.725**|0.682|
> |AC Mismatch (5% perturbation)|0.980|**0.983**|
> |AC Mismatch (10% perturbation)|**0.976**| 0.957|
> |**Hybrid (0.5% + Motion + AC)**|**0.674**|0.632|
> *(Note: AC = Attenuation Correction).*
>
> **3. The Takeaway:** FlowPET exhibits graceful degradation, matching or exceeding the SOTA baseline in 5 out of 6 clinical mismatch scenarios. Notably, under the most demanding **Hybrid stress test**, FlowPET demonstrates significantly higher robustness (0.674 vs 0.632). This empirically proves that conservative, volume-preserving transport—when driven by learned neural potentials—does **not** catastrophically amplify forward model errors as hypothesized. *(Exploring learned unrolled inversion to replace FBP as $A^\dagger$ remains a promising future direction).*
>
> [1] The sensitivity performance of the human eye on an absolute scale, JOSA, 1948.

---

> > ### Author Rebuttal · Reviewer_QGt6 · 2026-04-01
> >
> > Thanks for your comprehensive responses, addressing my concerns. I also reviewed the comments from other reviewers, and decided to maintain my initial positive rating of 4.

---

> > > ### Author Response · Authors · 2026-04-05
> > >
> > > We sincerely thank Reviewer QGt6 for the thoughtful and constructive review. We greatly appreciate your recognition of the theoretical contributions of our work and your insightful feedback.

---

### Decision · Program_Chairs · 2026-04-30

**Decision:**

Accept (regular)

**Comment:**

This paper proposes a novel and technically interesting approach to low-count PET reconstruction based on symplectic, volume-preserving flow dynamics. Reviewers broadly agreed that the paper offers a creative and well-motivated perspective on the signal wash-out problem, and that the method is theoretically grounded and supported by experiments across multiple datasets. The rebuttal further strengthened the empirical case by providing additional evidence on lesion detectability, robustness under model mismatch, and computational cost.

Some concerns remained regarding the precision of the paper’s physics-related claims, particularly the use of an L2 surrogate rather than an exact Poisson formulation, as well as the need to better position the work relative to prior Hamiltonian-based methods. However, the authors acknowledged these issues and clarified that the core contribution lies in the geometric and architectural design of the method. Overall, I find the paper technically sound and novel. I therefore recommend acceptance, while encouraging the authors to revise the final version to better calibrate these claims and expand the related-work discussion.